# Defect passivation in methylammonium/ bromine free inverted perovskite solar cells using charge-modulated molecular bonding

Dhruba B. Khadka [1] ✉, Yasuhiro Shirai [1] ✉, Masatoshi Yanagida [1], Hitoshi Ota [2], Andrey Lyalin [3,4] ✉, Tetsuya Taketsugu [4,5] & Kenjiro Miyano [1]

Molecular passivation is a prominent approach for improving the performance and operation stability of halide perovskite solar cells (HPSCs). Herein, we reveal discernible effects of diammonium molecules with either an aryl or alkyl core onto Methylammonium-free perovskites. Piperazine dihydriodide (PZDI), characterized by an alkyl core-electron cloud-rich-NH terminal, proves effective in mitigating surface and bulk defects and modifying surface chemistry or interfacial energy band, ultimately leading to improved carrier extraction. Benefiting from superior PZDI passivation, the device achieves an impressive efficiency of 23.17% (area ~1 cm²) (low open circuit voltage deficit ~0.327 V) along with superior operational stability. We achieve a certified efficiency of ~21.47% (area ~1.024 cm²) for inverted HPSC. PZDI strengthens adhesion to the perovskite via -NH₂I and Mulliken charge distribution. Device analysis corroborates that stronger bonding interaction attenuates the defect densities and suppresses ion migration. This work underscores the crucial role of bifunctional molecules with stronger surface adsorption in defect mitigation, setting the stage for the design of charge-regulated molecular passivation to enhance the performance and stability of HPSC.

Exceptional optoelectronic properties of halide perovskite (HP) have hiked the power conversion efficiency (*PCE*) of halide perovskite-based solar cells (HPSCs) over 26.1%, approaching to Shockley–Queisser limit[1,2]. A deluge of experimental efforts on stoichiometric engineering, crystallinity improvement, interface passivation, and carrier transport engineering has been used in the course of rapid progress[3,4]. However, HP films are still prone to degradation under external factors (such as thermal/humidity stress, oxygen, and light) and intrinsic phenomena[5,6]. These deleterious characteristics have exerted big challenges to practicality. Indeed, the HPSC degradation stems from deleterious surface chemistries on the surface of HP film or at the

device interface[7]. To address these detrimental defect chemistries, molecular passivation has been of great interest in improving the *PCE* and device stability[8,9].

Methylammonium (MA)-based HPs with a combination of various cations and halides have been extensively used in the state-of-the-art HPSCs[10-13]. However, MA is prone to be released and decomposes when exposed to elevated temperatures and humid conditions, which poses a persistent concern for device stability[14-16]. In recent years, MA-free formamidinium (FA)-based HP has garnered significant attention due to its higher thermal stability, enhanced moisture resistance, better absorbance in the near-infrared region, and tolerance to varying

[1]Photovoltaic Materials Group, Center for GREEN Research on Energy and Environmental Materials, National Institute for Materials Science (NIMS), 1-1 Namiki, Tsukuba, Ibaraki 305-0044, Japan. [2]Battery Research Platform, Research Center for Energy and Environmental Materials (GREEN), National Institute for Materials Science (NIMS), Namiki 1-1, Tsukuba 305-0044, Japan. [3]Research Center for Energy and Environmental Materials (GREEN), National Institute for Materials Science, Namiki 1-1, Tsukuba 305-0044, Japan. [4]Institute for Chemical Reaction Design and Discovery (WPI-ICReDD), Hokkaido University, Sapporo 001-0021, Japan. [5]Department of Chemistry, Faculty of Science, Hokkaido University, Sapporo 060-0810, Japan. ✉e-mail: KHADKA.B.Dhruba@nims.go.jp; SHIRAI.Yasuhiro@nims.go.jp; lyalin@icredd.hokudai.ac.jp

processing conditions[17,18]. However, it forms a photoinactive "yellow phase" at room temperature in pristine formamidinium (FA)- perovskite due to the relatively large size of FA. Alkali or organic cations have been introduced into the FA-perovskite film to promote the growth of the photoactive black phase at low-temperature crystallization. For example, Saliba and colleagues have reported the formation of a remarkably crystalline photoactive phase of FA-HP, achieved by introducing Rb and Cs ions without using MA and Br and subjecting the material to annealing temperatures not exceeding 100 °C. They have achieved a competitive *PCE* of 20.35% with improved stability[19]. Similarly, Seok and co-workers demonstrated an impressive *PCE* of 24.4% using alloyed FA-perovskite with organic cation in the normal device configuration[20]. It has documented the formation of a highly crystalline α- phase by incorporating methylenediammonium dichloride as a divalent organic cation with an ionic radius equivalent to FA which induced a stronger ionic interaction of its divalent state in FA-HP.

In recent scenarios, several studies have focused on the versatile molecular passivation strategy aimed at mitigating various intrinsic point defects in the HP film[4,21,22]. Surface or bulk areas can lead to the formation of under-coordinated $Pb^{2+}$ ions, A-site vacancies/interstitial defects, and halide vacancies, causing recombination and performance loss[22–25]. The molecular passivator with multifunctional derivatives consisting of amine[26,27], Lewis acids/bases[28,29], supramolecules[30–32], ionic polymer[9,33], etc. have been used for mitigation of defect chemistry in the HPSCs. The ammonium-containing functional additives with alkyl or aryl halide or pseudohalide counterparts have demonstrated a significant enhancement in *PCE* and operational stability[34–41]. The additive having stronger adsorption on the HP surface modulates the grain nucleation and growth or alleviates defect chemistry at interface and bulk, which is crucial for a highly efficient and stable device[22,42]. Additionally, some functional molecules eliminate molecular iodine present within the perovskite and suppress both iodine and Pb-related deep trap sites[43,44]. Importantly, the introduction of molecular functional derivatives of diammonium halide into alkyl or aryl core, serving as additives or passivation, has also exhibited noteworthy enhancements in both the *PCE* and stability of HPSCs[45–49]. For example, Wakamiya and co-workers have demonstrated the universality of surface treatment with ethylenediammonium diiodide by wet and dry deposition methods on perovskite surfaces, achieving a decent *PCE* and device stability in various perovskite systems[45,50]. Liu and co-workers introduced a highly electronegative fluorine molecule; Cobalt (II) hexafluoro-2,4-pentanedionat for interface passivation of MA-free HPSCs, resulting in the mitigation of defect chemistries and enhancing hole-transport kinetics[51]. This modification demonstrated a remarkable PCE of 24.64% (normal device structure) as a consequence of strong molecular interaction with the surface charge modulated passivation. Typically, inverted devices exhibit lower efficiency levels than conventional structures. Inverted HPSCs with an inorganic hole transport layer have gained significant interest due to their potential for superior stability and compatibility with tandem solar cells[52,53]. An inverted HPSC with MA-free HP mixing Tris(chloromethyl) ammonium iodide using $NiO_x$ nanoparticle as HTL reported a decent certified PCE of 23.2% with a small active area (0.04 cm$^2$)[42]. Despite achieving an impressive *PCE* of record level in small-area HPSCs, there remains a substantial *PCE* disparity between small and large-area PSC devices[4,54]. Therefore, the fabrication of highly efficient inverted HPSC with a large area continues to pose a challenge. Despite a variety of functional molecules being wielded for the optimization of crystal growth and defect passivation in MA-free HP, the influence of adjusting surface charge via bonding interactions is often overlooked.

In this work, we report on passivation strategy through bond/charge regulated defect passivation by introducing bifunctional molecules with an aryl core (1,4-phenylenediamine dihydrodide (PEDAI)) or alkyl core (piperazine dihydriodide (PZDI)) onto the MA/Br-free 3D-HP film ($FA_{0.84}Cs_{0.12}Rb_{0.04}PbI_3$) in inverted HPSCs. We demonstrate that the different molecular properties lead to distinct differences in the film growth, material distribution, and device characteristics. The bifunctional molecular passivation significantly affects the film morphology and surface chemistries by their basicity and adsorption energy. Thus, the PZDI additive effectively quenches the surface or bulk defects in the HP film resulting in a longer carrier lifetime and better interface quality. Consequently, the HPSCs with PZDI treatment resulted in an enhanced large area (>1cm$^2$) device performance from 19.68 to 23.17% (inverted configuration) with an increase in the device parameters and reduced $V_{OC}$ deficit of 0.327 V with superior device stability under thermal and moisture stress. Density functional theory (DFT) calculations show that the alkyl core amine has stronger bonding interaction with uncoordinated $Pb^{2+}$ and iodine traps. This method is equally effective in improving device performance in narrow and wide bandgap HPSC systems. The work gets insight into the device characteristics with synergetic effect in the surface chemistry, elemental distribution, photophysics, and defect profile.

## Results and discussion
### Surface passivation and film growth characterization
Figure 1a illustrates the chemical structures of diammonium iodide molecules (DIMs) with aryl and alkyl cores, respectively, employed as bonding/charge-regulated molecular passivation. The optimized structures of the free molecules, namely PEDAI ($C_6H_8N_2$*2HI) and PZDI ($C_4H_{10}N_2$*2HI), were obtained using Gaussian software at the B3LYP/def2TZVP level of theory. The bond lengths and Mulliken charges for corresponding molecules are shown in the adjoining figure. These molecules exhibit distinct characteristics in terms of bond length and Mulliken charge distribution, which are crucial for effectively suppressing charge defects in the perovskite film. Notably, considering the charge cloud induced in iodine and nitrogen atoms within the PEDAI and PZDI molecules, the PZDI induces stronger adsorption resulting in effective molecular interactions with uncoordinated Pb atoms and charge defects within the perovskite film. The DFT calculation section provides a comprehensive discussion of the theoretical aspects.

To test the effectiveness of DIMs in a perovskite device, these molecules were introduced on the 3D-HP film as an interfacial passivation layer (IPL) as shown in schematics (Fig. 1b, c). The surface morphology of MA/Br-free HP film (treated without and with PEDAI or PZDI) was studied with scanning electron microscope (SEM) measurement (Fig. 1d–f). One can notice a slight change in the grain size with an overlayer surface grown on the film. The HP film with PEDAI passivation grows with unevenly distributed small crystallite. It suggests that the DIM with an alkyl core has beneficial surface coverage compared to the aryl core. The film with PZDI treatment forms well-covered surface and grain boundaries which is propitious for the elimination of localized defects in perovskite film[8,22].

We collected X-ray diffraction (XRD) results of control and surface-treated perovskite films to study the crystal growth. XRD results of the PEDAI or PZDI-treated HP films with reference to the control film are displayed in Fig. 1g and Fig. S1a, b. The PZDI-treated HP films show a dominant (110) characteristic diffraction peak of the α-phase of FA- HP. It suppresses the δ-$Cs/RbPbI_3$ phase and residual $PbI_2$. Moreover, additional peaks at 2θ < 10° are dominantly grown on the HP films with a higher content of DIM indicating the evolution of a 2D phase of PEDAI and PZDI interacting with $PbI_2$[48]. The characteristic XRD peak at lower 2θ on the HP film with PEDAI treatment grows with higher intensity indicating higher tendency for the formation of 2D phase compared to alkyl counterpart. This result is parallel to the

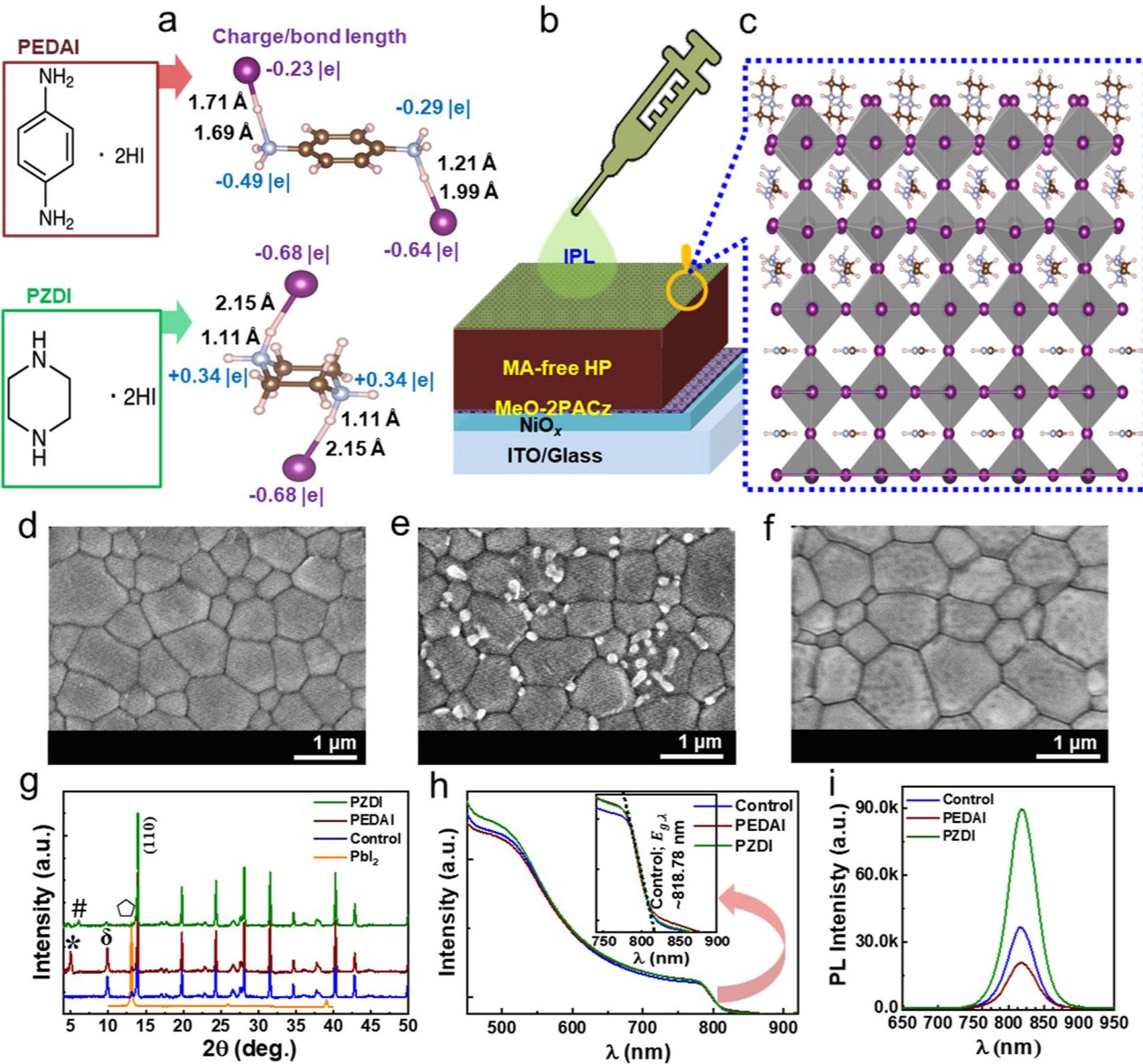

**Fig. 1 | Schematic of surface treatment and film growth. a** Chemical structure, bond length, and Mulliken charges with summed H (iodine and nitrogen atoms) (PEDAI and PZDI). **b** Surface passivation of MA-free ($FA_{0.84}Rb_{0.04}Cs_{0.12}PbI_3$) 3D-HP film. **c** Illustration of interfacial interaction of diammonium iodide molecular passivator and HP surface. **d–f** SEM images of HP films; without passivation, with PEDAI and PZDI passivation. **g** XRD patterns. **h** Absorption spectra. **i** PL spectra without and with IPL treatment on the HP films.

formation of low-dimensional perovskite on Sn-Pb 3D perovskites through surface treatment with piperazine-based amine[49].

Figure 1h depicts the absorption spectra of the control and DIM-treated HP films, with varying concentrations (as shown in Fig. S1c, d). Notably, there is no significant difference observed at the band edge of the HP film as evidenced by the insets in Fig. 1h and Fig. S1c, d. The PL spectra (Fig. 1i) of HP with PZDI demonstrated an intensified peak while PEDAI suppressed it. The variation in PL intensity among the HP films treated with different concentrations of DIM (Fig. S2) aligns with an improvement in opto-physical quality. One can see a minimal shifting of the PL characteristic peak (~817–818 nm). This suggests that these additives did not incorporate into the 3D-HP lattice which is also consistent with XRD patterns.

To gain deeper insights into the formation of the 2D phase, we prepared HP film by mixing DIM in the perovskite precursor solution, as depicted in Fig. S3a$_1$-b$_1$, and powder crystal by mixing ($PbI_2$ and PEDAI or PEDAI). XRD results revealed that the HP films with DIM mixing and powder crystal grow without any remanence of the $PbI_2$ peak suggesting a strong tendency for the formation of a new crystal phase. Besides that, XRD patterns (Fig. S3a$_2$-b$_2$) demonstrated a more intensified XRD peak at 2θ<10° for the HP film prepared with mixed DIM compared to the surface treatment alone. Importantly, the HP film with mixed PEDAI was found to be grown with much higher XRD peak intensity at ~4.8° indicating a more preference towards 2D phase formation. This observation was further supported by SEM images as displayed in Fig. S3a$_3$-b$_3$ which depicted nano-sheet-like features in the HP mixed containing PEDAI, underscoring the presence of PEDAI-based 2D phase. This observation is consistent with the small crystallite observed in the PEDAI passivated film. Interestingly, in contrast to the PEDAI-mixed scenario, the HP film with mixed PZDI exhibited growth as an overlayer on crystal grains or at grain boundaries, rather than forming a flake structure that closely resembled that of the passivated film. To confirm the crystal structure of the 2D phase with PZDI in HP film, single crystals were grown by mixing 2:1 and 1:1 molecular

ratios of PZDI and $PbI_2$ using the reported antisolvent vapor-assisted crystallization method as described in Fig. S4. We obtained single crystal (2D phase) of composition $(PZDI)_2(PbI_4)_2$ (orthorhombic) (Supplementary data 1, CCDC 2311444) and $(PZDI)_3Pb_2I_7$ (monoclinic) (Supplementary data 2, CCDC 2311446) with DMSO complex confirmed by single crystal XRD analysis. The results (Supplementary Fig. S5 and Table S1) show the simulated XRD patterns and corresponding crystal unit and packing fraction. Although the growth condition is completely different from the film preparation, the XRD data of PZDI-contained HP film are found to be close to the synthesized single crystal data. This confirmed the formation of small crystallite on the surface of HP film with PZDI treatment.

Furthermore, the PL spectra of respective HP films (Fig. S3c) also showcased distinct PL peaks at higher energy regimes suggesting the existence of DIM-based 2D phases. Notably, the PL characteristic features of respective films are parallel to XRD patterns. These results collectively suggest the HP films with PEDAI tend to favor 2D phase formation rather than forming a fully covered 3D-HP surface. It is believed that these characteristic differences could have a significant effect on the device's performance.

## Photovoltaics performance and photophysics

To investigate the effect of surface treatment using DIM with aryl or alkyl core on the photovoltaic properties, we fabricated HPSCs with inverted device configuration as depicted in Fig. 2a. A typical cross-sectional image of a complete device is displayed in Fig. 2b. The current density-voltage (J-V) curves for the best HPSCs without and with PEDAI or PZDI treatment are shown in Fig. 2c with a large device area of $\approx 1\,cm^2$ (Fig. S6) The J-V characteristics with varying concentrations of DIM are given in Fig. S7. The device parameters have been summarized in Table 1 (Tables S2–S3). The HPSCs with PZDI-treated HP (1 mg/ml) demonstrated champion PCE of ~23.17% with negligible J-V hysteresis. However, contrary to our expectation, the PCE of the PEDAI passivated device rolls off with reduced $J_{SC}$ and FF with reference to the control device as given in Table 1. A decrease of $J_{sc}$ and an increase of $V_{oc}$ (both small) seem to be typical behavior of slight passivation of interface defects. It suggests that PEDAI has an interface passivation effect to some extent. These results corroborate that the amine in DIM with aryl and alkyl core greatly affects the device results. It is a consequence of the chemical interaction of the HP with DIM that drives the film quality impacting morphology, surface chemistry, and defect profile. It is known that PEDAI has a delocalized lone pair of electrons of nitrogen atoms while that in PZDI is localized. The nitrogen site in DIM enhances the surface adhesion on the HP film which is favorable for defect mitigation. This could result in a stark characteristic difference in HPSCs. We will discuss more insight in the succeeding paragraph accounting for these aspects.

Figure 2d presents the PCE statistics of the control device and that with PEDAI or PZDI treatments (Figs. S8–10 and Tables S2–S3). For PZDI, the device performance is found to be improved with an increase in all device parameters. However, the device with a higher concentration of PZDI rolls off with a lower $V_{OC}$ and FF. In contrast, the HPSCs with PEDAI treatment are inferior to the control device and the device with a higher concentration of PEDAI further deteriorates by dropping $V_{OC}$ and FF like the PZDI case. These inferior device parameters with higher PEDAI or PZDI content could be due to the accumulation of 2D HP phase on the surface unevenly. This observation is in line with other reports[32,48].

We validated HPSCs with MA/Br-free HP film with PZDI treatment of PCE $\approx 21.47\%$ (area $\approx 1.024\,cm^2$) under standard conditions (accredited independent photovoltaic test laboratory, AIST PV Lab, Japan). The official certified data is given in Fig. S11. Our certified PCE of the champion device has a record-level device efficiency for inverted p-i-n configuration of HPSCs with MA/Br-free HP for a large area of $>1\,cm^2$. For comparative evaluation, the few certified device reports with an area $>1\,cm^2$ are tabulated (Table S4), where our champion HPSC compares favorably amongst the reports.

The external quantum efficiency (EQE) data for HPSCs without and with DIM passivation are shown in Fig. 2e. The EQE response for the device with PZDI presents a better spectral response in the absorption energy band range of the HP layer ($\lambda > 450\,nm$) and interfacial regime ($450 > \lambda > 330\,nm$)[55]. It is attributed to the betterment in the bulk and interface quality of PZDI-treated HPSC. Note that the integrated current values from EQE spectra are 23.04, 22.68, and 24.08 $mA/cm^2$ for the control and PEDAI or PZDI-treated HPSCs, which are in the range of the $J_{SC}$ of respective devices. We also calculated the bandgap ($E_g$) of the HP absorber layer from EQE analysis ($E_g^a$-1.513, 1.516, and 1.515 eV for the control, PEDAI, and PZDI; Fig. S12a–c). They are in close agreement with the $E_g$ obtained from the absorption spectra (Fig. S9d–f) and PL spectra (Fig. 1i).

To explore the characteristic insight, we analyzed the photo response of the HPSCs. Figure 2f presents the $V_{OC}$ variation with logarithmic of light intensity (ln(I)). The slopes are estimated to be 1.40, 1.34, and 1.16 $k_BTq^{-1}$ for the control, PEDA, and PZDI-passivated devices, respectively. A device with a higher slope signifies more charge recombination at open circuit conditions. It suggests that the HPSCs with PZDI experience reduced trap-assisted recombination which ameliorates the device performance. We recorded the TPV response as displayed in Fig. 2g by triggering $V_{OC}$ with transient photo illumination. The TPV decay signal analysis reveals a longer carrier lifetime for the PZDI-treated device (12.34 $\mu s$). While the device with PEDAI (7.62 $\mu s$) shows a slight increase in carrier lifetime compared to the control device (6.22 $\mu s$). It suggests that the PZDI treatment passivates the defect in the HP film with propitious surface chemistry and mitigating defect.

To understand the carrier lifetime, we measured the time-resolved photoluminescence (TRPL) spectra (Fig. 2h) and fitted them with a bi-exponential decay equation[48]; $I(t) = A_0 + A_1 e^{-\frac{(t-t_0)}{\tau_1}} + A_2 e^{-\frac{(t-t_0)}{\tau_2}}$, where $A_0$ is a constant for the baseline offset, $A_1$ and $A_2$ are the relative amplitude. The decay time, $\tau_1$ and $\tau_2$ accounts for the nonradiative recombination at the interface and radiative recombination at the bulk layer[56]. The HP film with PZDI treatment shows a significantly longer lifetime ($\tau_1 \approx 331\,ns$ and $\tau_2 \approx 2285\,ns$) compared to that of the HP with PEDAI ($\tau_1 \approx 85\,ns$ and $\tau_2 \approx 783\,ns$) or control ($\tau_1 \approx 98\,ns$ and $\tau_2 \approx 678\,ns$). Interestingly, the PEDAI-treated film shows only a small difference in carrier lifetime compared to the control film. It corroborates that PEDAI is not as effective as PZDI for the attenuation of a deleterious defect in the HP film. These results indicate the surface treatment with PZDI is propitious for defect passivation due to stronger localized nitrogen bonding in HP film and hence leads to the superiority of device performance.

## Modulation of surface chemistry and interface

To understand surface energy, we measured ultraviolet photoelectron spectroscopy (UPS). The cutoff energy corresponding to the work function ($\phi$) (Fig. 3a) and the onset energy ($E_i$) (Fig. 3b) calculated from the UPS results. The band structure has been constructed by combining with optical bandgap and UPS result (Fig. S13). The values of $\phi$ and $E_i$ are found to be slightly increased with DIM treatment. The results demonstrate a downshift of $E_V$ (by 0.287 or 0.278 eV) and $E_C$ (0.283 or 0.276 eV) levels for PEDAI or PZDI-treated film. It modulates the band alignment at the HP/$C_{60}$ interface[27,36]. Indeed, the interfacial band alignment induced by DIM treatment is beneficial for effective carrier transport resulting in better device performance. Although the PZDI or PEDAI shows a similar effect in surface energy, there is a significant improvement in device parameters for the device with PZDI treatment. It suggests that the surface energy modification by PEDAI has only a minimal effect on device performance.

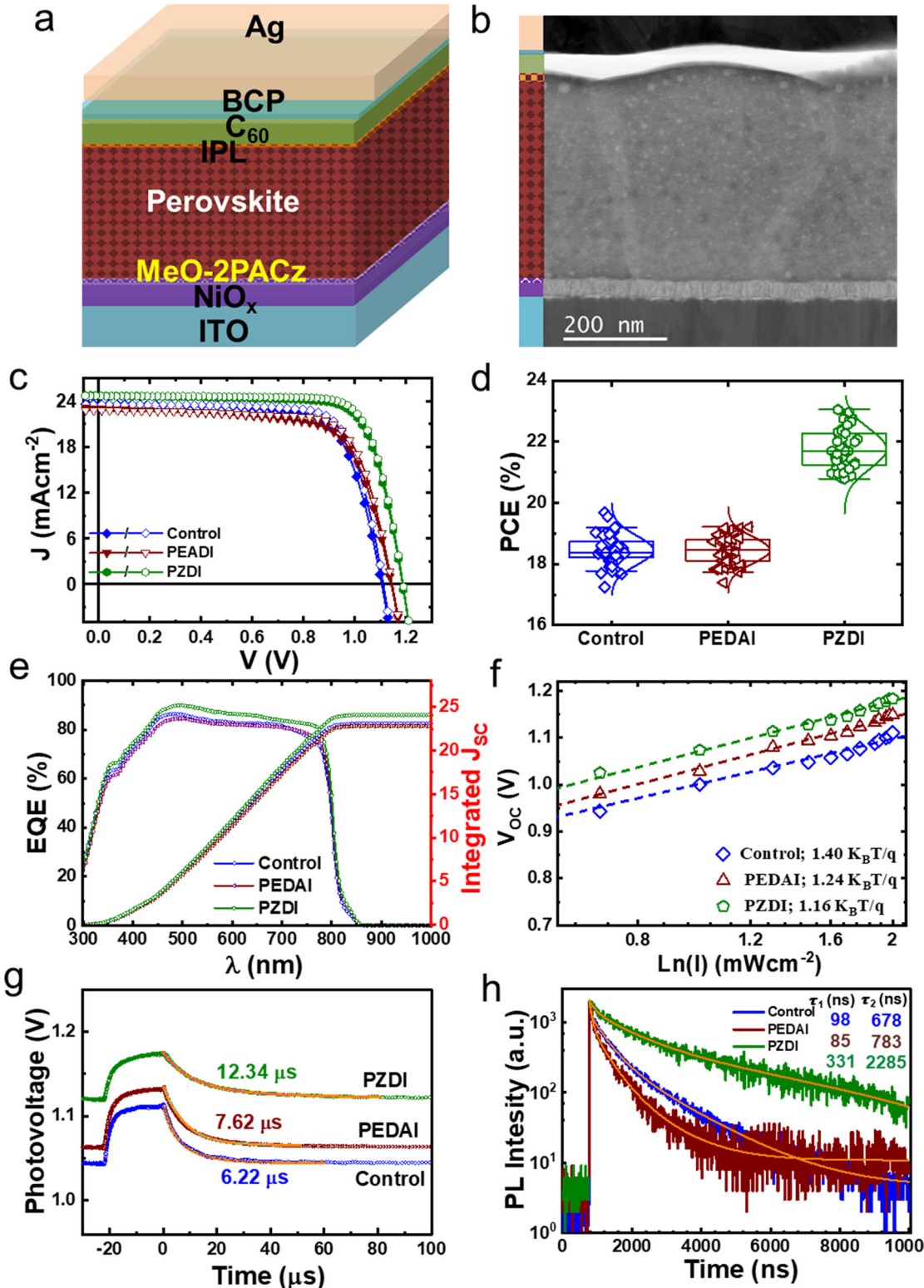

**Fig. 2 | Photovoltaic characterization of the HPSCs with surface treatment.**
**a** Schematics of the device structure. **b** STEM-cross-sectional images of devices.
**c** J-V characteristics of the control and IPL treatment (best IPL content; 1mg/ml);
(open/filled symbols (forward/reverse) scan direction). **d** Statistical box for *PCE* of
the HPSCs without and with surface passivation (PEDAI or PZDI). These data consist
of 50 devices from 6 batches. **e** *EQE* spectra of devices. **f** Light intensity vs. $V_{OC}$
plot. **g** Transient photovoltage (*TPV*) decay curves of the respective devices. **h** *TRPL*
decay spectra for corresponding films.

To explore surface chemistry, we carried out X-ray photoelectron spectroscopy (XPS) measurements. In the C *1s* XPS core (Fig. 3c), the binding energies centered at ~284.4, 286.2, and 287.4 eV are assigned to C-C/C=C, C-N-C, and N = C-N, respectively. The C-N-C characteristic peak for PZDI-treated film indicates its dominant interaction with the

HP surface. The N *1s* XPS cores for the corresponding film (Fig. 3d) also indicate the respective chemical binding characteristics. The surface passivated film shows a small shift of the XPS characteristic core of Pb *4f* and I *3d* (-0.11 and 0.18 eV for PEDAI and 0.16 and 0.29 eV for PZDI; respectively, Fig. S14) towards higher binding energy. It indicates a

**Table 1 | Solar cell data and $V_{OC}$ deficit of the HPSCs with MA/Br-free perovskite (without and with surface treatment (PEDAI or PZDI))**

| Condition | $E_g^a$ (eV) | Scan | $J_{SC}$ (mAcm$^{-2}$) | $V_{OC}$ (V) | FF | PCE (%) | $V_{OC}$ deficit ($E_g^a/q$ -$V_{OC}$) (V) |
|---|---|---|---|---|---|---|---|
| Control | 1.513 | F | 23.62 | 1.104 | 0.717 | 18.69 | 0.402 |
| | | R | 23.56 | 1.112 | 0.750 | 19.65 | |
| PEDAI | 1.517 | F | 22.98 | 1.142 | 0.703 | 18.46 | 0.381 |
| | | R | 22.79 | 1.145 | 0.736 | 19.21 | |
| PZDI | 1.515 | F | 24.76 | 1.185 | 0.772 | 22.65 | 0.327 |
| | | R | 24.78 | 1.188 | 0.787 | 23.17 | |

$V_{OC}$ deficit calculated with reference to $E_g^a$ extracted from EQE data. The letters: F and R refer to the scan directions.

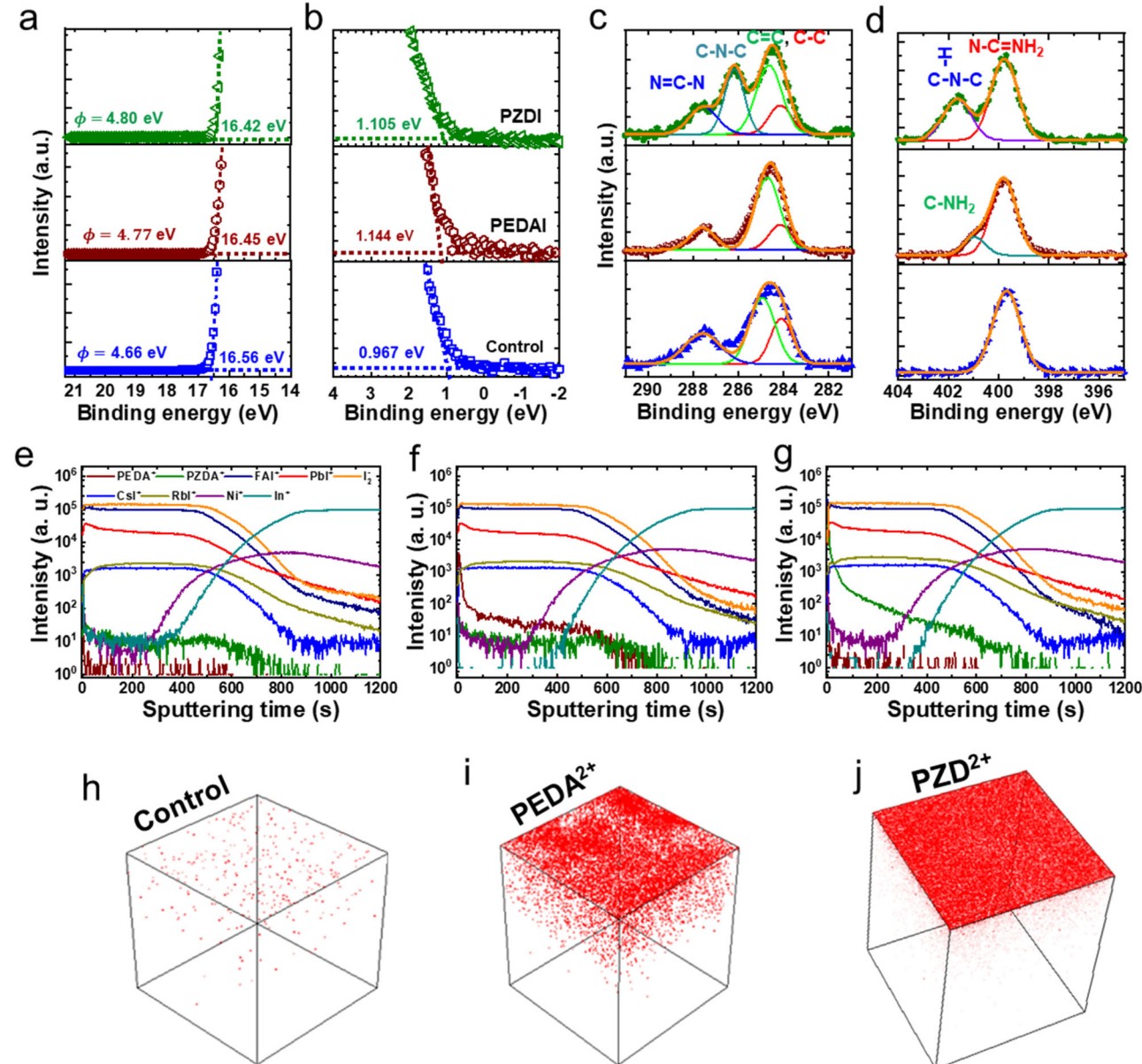

**Fig. 3 | Characterizations of surface chemistry and elemental distribution.**
**a**, **b** UPS spectra of the HP films; control, PEDAI, and PZDI treatment. **c**, **d** XPS-spectra analysis; C-*1s* core and N-*1s* core. **e**–**g** ToF-SIMS depth profiles of control, PEDAI, and PZDI passivated HP films. **h**–**j** Reconstructed 3D maps; the distributions of passivated molecules in HP film. There are selected ionic species; ITO (In$^+$), NiO$_x$ (Ni$^+$), HP control (FAI$^+$, PbI$^+$, CsI$^+$, RbI+) and with PEDAI (PEDA$^{2+}$) or PZDI (PZD$^{2+}$).

stronger ionic bonding induced on the film surface with PZDI treatment. The Cs $3d$ and Rb $3d$ core levels demonstrate almost similar spectral features indicating only a weak interaction with the passivating molecule. This surface analysis implicates that the bifunctional surface passivator establishes stronger interaction with nitrogen bonding to uncoordinated $Pb^{2+}$ or Iodine antisites[48,57]. The functionality of the PZDI molecule is superior due to higher electron density in the vicinity of the N-atom compared to PEDAI with an aryl core.

Furthermore, we conducted a cross-sectional transmission electron microscopy (TEM) measurement to investigate the interfacial structure formed with the 3D HP host. A set of Fig. S15 illustrates the scanning transmission electron microscopy (STEM) images of both the control and PEDAI or PZDI-treated HPSCs, revealing a noticeable feature at the HP/$C_{60}$ interface (Fig. S15b$_{3-4}$ and S15c$_{3-4}$). High-resolution TEM (HR-TEM) images at the interface of the PEDAI or PZDI-treated device display a wide interplanar d-spacing (Fig. S15b$_4$-c$_4$), providing evidence for the presence of a 2D phase either at the interface or buried within the 3D HP host HP. One can see an unevenly distributed 2D phase not only at the HP/$C_{60}$ interface but also in the subsurface bulk for PEDAI-treated devices which could be detrimental to device quality. We found that both molecules, PEDAI and PZDI, modify the surface or grain boundary chemistries and the interfacial band structure by virtue of their molecular functionalities. This finding aligns with previous studies on interface passivation[39,58].

To explore the spatial distributions of molecular passivator, time-of-flight secondary ion mass spectrometry (ToF-SIMS)[32,45,59] was used to track the ionic distribution of PEDA$^{2+}$ and PZD$^{2+}$ in the HP. The characteristic ionic species are shown in Fig. 3e–g. It shows an identical ionic distribution in the perovskite bulk. The characteristic signals from PEDAI or PZDI are found to be significantly higher on the surface with a deep gradient to the bulk (Fig. S16). The 3D maps (Fig. 3h–j) demonstrate that the PEDA$^{2+}$ and PZD$^{2+}$ cations introduced by surface treatment are mainly distributed on the HP top surface. This observation is analogous to the report by the Wakamiya group in ethylenediammonium iodide-treated Sn-Pb mixed HP film[45]. Importantly, the PZD$^{2+}$ additive shows uniform surface coverage with negligible bulk diffusion. In contrast, the PEDA$^{2+}$ ions are found to have uneven distribution on the top surface with enriched bulk diffusion compared to the PZDI case. These results are analogous to the surface feature of SEM images of respective films. This observation corroborates that the PZDI or PEDAI mainly passivates the surface defect density by its functional characteristics to modify the surface chemistry of the pristine perovskite layer.

Moreover, we also delved into the analysis of residual stress in the respected devices following surface treatment by examination of XRD spectra using the $2\theta-\sin^2\psi$ method[60], as detailed in Fig. S17. We selected the (012) plane at 31.6° as the focal point for analysis due to its ability to provide grain information and its diluting orientation effect in the linear relationship of $2\theta$-$\sin^2\psi$. The data illustrate how the scattering peaks progressively shift to the left as the $\psi$ angle varies from 10 to 60° at different levels. The stress induced in perovskite can be calculated by fitting the plots ($2\theta$-$\sin^2\psi$). Notably, a negative slope indicates the presence of tensile stress within the perovskite film. A lower negative slope was observed for PZDI treatment compared to PEDAI and control device which aligns with the other reports[60,61]. Moreover, the GIXRD analysis confined to the (012) plane (Fig. S18) was also analyzed accounting for the penetration depth corresponding to the angle of the grazing incident. It revealed a distinct effect with an increasing depth profile suggesting the modification of the surface, grain boundaries, and the bulk of HP film through the surface treatment. This could have a crucial role in strain attenuation. It corroborates that the surface treatment passivates the defect on the surface or at grain boundaries to some extent which results in lower strain in the film. Thus, it is suggested that the surface treatment using PZDI with stronger -NH functionality regulates the HP film by mitigating the residual strain which is crucial in influencing carrier dynamics, device results, and device stability as observed in this work.

## Effect of surface treatment on defects

To get insight into the defect densities, we investigate the admittance spectroscopy of HPSCs with surface treatment. Mott-Schottky (M-S) plot and carrier profile ($N_{CV}$) were extracted from capacitance-voltage (C-V) data[62-65]. M-S plots (Fig. 4a) exhibit fully depleted curves for $V >$ diffusion potential ($V_D$) suggesting intrinsic characteristic junction. One can see a slight hysteresis in the M-S curve near $V_D$ for the control device which almost disappeared for the surface-passivated device. It is attributed to the reduction in ionic polarization at the interface. The $V_D$ value for the PSC with PZDI (1.148 V) is greater than PEDAI (1.065 V) and control device (0.991 V) which is parallel to the $V_{OC}$ of the respective device. The carrier profile ($N_{CV}$) (Fig. 4b) extracted from C-V data analysis comprises the free carrier and defect density. It showed a carrier distribution in a bulk ($N_{CV}^B$) in the range of $\sim3.46$–$6.94 \times 10^{15}$ cm$^{-3}$. The $N_{CV}^B$ with PZDI is slightly lower by some fraction. The carrier profile at the edge accounts for the interface defect density profile ($N_{CV}^{IF}$ $\sim15.26 \times 10^{17}$, $-11.56 \times 10^{17}$, and $\sim2.43 \times 10^{17}$ cm$^{-3}$ for control, PEDAI, and PZDI treated devices, respectively). The interface defect density is suppressed by 6 times for the PZDI-passivated device. It corroborates that the PZDI effectively attenuates the recombination centers leading to the improvement in the device parameter.

For the quantitative analysis of the defect profile, we investigated thermal admittance spectroscopy, an effective technique for estimating optoelectronic properties; the defect level and defect density to thin film solar cells HPSCs[66,67], chalcogenide solar cells[68], and organic solar cells[69]). Figure 4c shows the capacitance-frequency (C-f) spectra measured at room temperature (under dark). All device reveals a plateau regime (1 to 100 kHz) with a slightly lower value for PZDI treated device that could stem from the HP accounting for defect density. Besides that, the lower frequency capacitance response is much steeper for the control device which is attributed to the interfacial charge accumulation or ionic polarization. It implicates a suppression of interfacial charge accumulation for the HPSCs with PZDI treatment.

Furthermore, we measured the temperature-dependent capacitance-frequency (C-f-T) spectra to analyze the defect density profile. The trap state ($E_t$) is calculated from Arrhenius plot[70] by analyzing the resonance frequency ($\omega_o$) obtained from the C-f-T analysis as given in Fig. S19. The Arrhenius plots (Fig. 4d–f) revealed shallower defect states in the PZDI-treated device ($E_{t3}, E'_{t3} \sim 0.154, 0.374$ eV) compared to the PEDAI-treated ($E_{t2}, E'_{t2} \sim 0.212, 0.408$ eV) or control device ($E'_{t1}, E'_{t2} \sim 0.241, 0.423$ eV). We calculated the defect density profiles (Fig. 4g–i) using the equation[70,71], $N_t(E_\omega) = -\frac{V_D}{qW}\left(\frac{\omega}{k_B T}\frac{dC}{d\omega}\right)$, where, $V_D$, $W$, $q$, and $\omega$ denote the diffusion potential, the space charge region width, elementary charge, and applied frequency, respectively.

We found that the integrated trap densities for the control device ($N_{t1}, N'_{t1} \sim 1.08 \times 10^{17}$, $9.89 \times 10^{16}$ cm$^{-3}$) are attenuated for the PEDAI ($N_{t2}, N'_{t2}$-$7.38 \times 10^{16}$, $8.49 \times 10^{16}$ cm$^{-3}$) or PZDI ($N_{t3}, N'_{t3}$-$3.22 \times 10^{16}$, $4.03 \times 10^{16}$ cm$^{-3}$) treated devices. These results are in the range of reported trap densities for the perovskite film ($10^{16}$–$10^{19}$ cm$^{-3}$)[27,72]. On other hand, the defect densities in our devices are more than $10^6$ than a single crystal ($10^{10}$ cm$^{-3}$) which calls for more effort to lower the trap densities to achieve superior film quality. From the comparative analysis, the trap densities ($N'_{t1}$) primarily assigned for defects in the bulk are decreased by ~2.5 times in the device with PZDI treatment or 1.65 times in the device with PEDAI treatment indicating the improved bulk quality of HP film. The shallower trap state profile ($N_{t1}$) is assumed to be defects at the surface or GBs in the HP film. These shallower defect densities are found to be significantly lowered in the PZDI or PEDAI-treated devices. These results consolidate that the PEDAI or PZDI stays on the film surface or diffuses into the bulk through the GBs to passivate the defect states due to molecular interaction with the

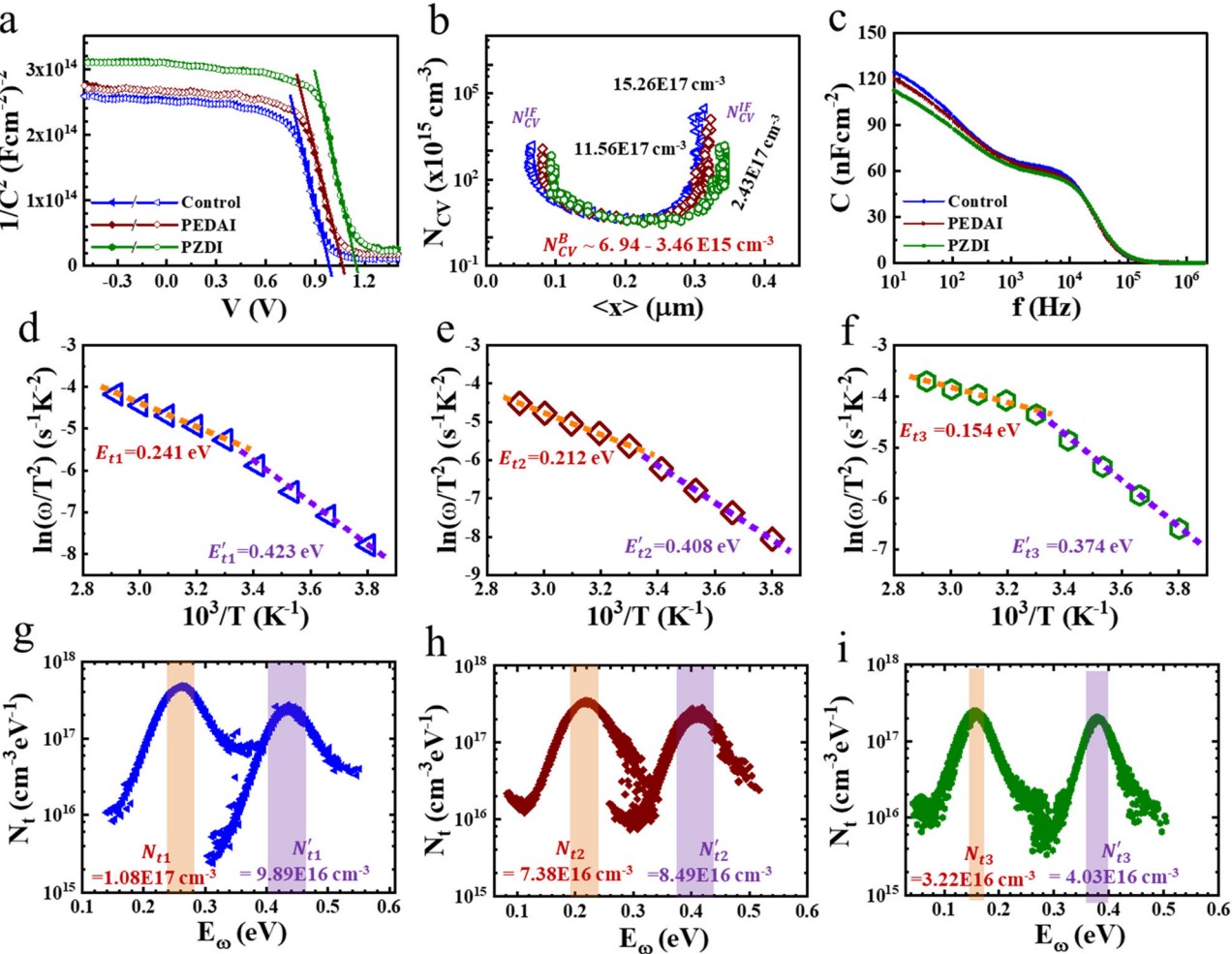

**Fig. 4 | Capacitance characteristics of HPSCs. a** Mott–Schottky plots (open/filled symbols forward/reverse scan direction). **b** Carrier distribution calculated from *C-V* data. **c** C-f response. **d**–**f** Arrhenius plots. **g**–**i** Defects profile ($N_t$).

characteristics of nitrogen terminals. We found that the PZDI passivation is rather efficient for mitigating the defect chemistries at the surface, GBs, and bulk compared to PEDAI. Thus, the capacitance spectra analysis well agrees with the advantageous properties of HP film induced with surface treatment as discussed in previous paragraphs.

## Theoretical insights on surface passivation

To elucidate the effects of the DIM passivator on the HP film, we conducted the first-principles calculations based on density functional theory (DFT). A detailed description of our computational and theoretical methods can be found in the Supplementary Information. The pseudo-cubic structure of $FAPbI_3$ was used as a model of the bulk structure (Fig. S20). The perovskite's surface was modeled by a 2 x 2 slab of $PbI_2$-terminated surface (001) with five $PbI_2$ layers and a vacuum region of ~25 Å (Fig. S21a). The corresponding total density of electronic states (DOS) calculated for the defect-free $PbI_2$ terminated surface is shown in Fig. S21b. In the case of full surface coverage, we accommodated two PEDAI (PZDI) molecules (Figs. S22 and S23), with the I atoms of PEDAI (PZDI) adsorbing atop the Pb atoms of the topmost $PbI_2$ surface layer, as illustrated in Fig. S24a, b. Both molecules are assumed a tilted orientation, with their N atoms forming a plane parallel to the perovskite surface. Our calculations revealed that PEDAI molecules exhibit a preference for forming a chain along the (100) direction, while PZDI molecules form a chain along the (010) direction. The interaction of PEDAI and PZDI with the surface induces a slight

distortion of the surface $PbI_2$ layers and causes a rotation of the FA molecules within the first and second $PbI_2$-FAI bilayers. This effect is particularly prominent in the case of PEDAI@$FAPbI_3$ (see Fig. S24a). It could also affect the distribution of passivating molecules during the film formation.

As one can see, adsorption of PEDAI or PZDI molecules does not introduce any defect states in the forbidden zone of defect-free perovskite (DOS for PEDAI and PZDI; Fig. S24c, d), slightly increasing the bandgap from 1.53 eV calculated for the pure $FAPbI_3$, up to 1.60 eV (1.59 eV) for the perovskite covered by PEDAI (PZDI) which is analogous to slightly higher surface band energy obtained from UPS analysis. Both DIM molecules are attached strongly to the defect-free surface, which protects the unstable surface without introducing any adverse electronic effects. In contrast, on the defective surface, these molecules show remarkable electronic functions.

Figure 5 shows the role of PEDAI/PZDI passivation of the defected ($I_{Pb}$ antisite) $PbI_2$-terminated surface of $FAPbI_3$ defined in the previous studies[25,32]. Indeed, the $I_{Pb}$ antisite defect results in the formation of an unoccupied defect state 0.1 eV above the Fermi level as well as some defect states in the middle of the forbidden zone (0.4–1.0 eV above the Fermi level), as it is seen from the analysis of the total DOS of the perovskite surface with $I_{Pb}$ defect, presented in Fig. 5d, e by black lines. The detrimental defect states can be effectively passivated with PZDI treatment. Thus, in the case of PZDI passivation, the density of the defect states is considerably reduced, slightly shifting down the bottom of the conduction band by 0.06 eV toward the Fermi level (Fig. 5e).

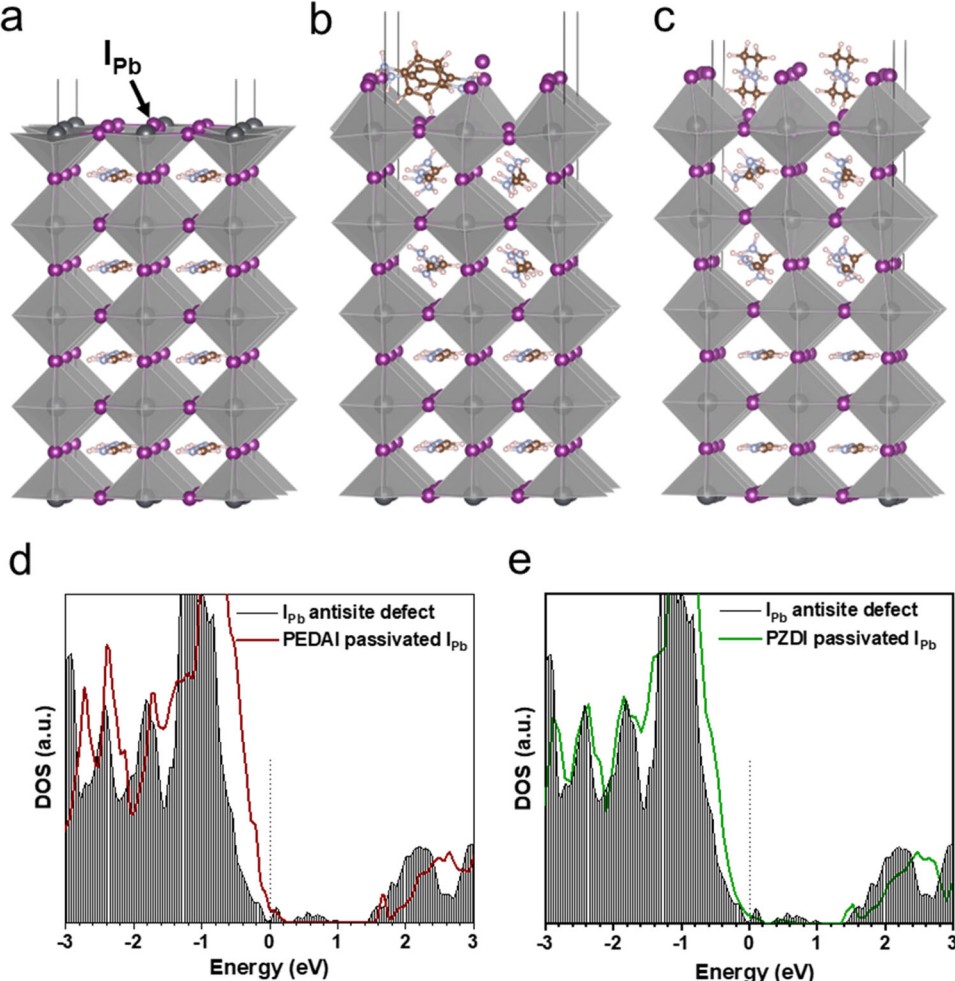

**Fig. 5 | DFT calculations of defect passivation. a** Optimized structures of $I_{Pb}$ antisite defect. **b, c** PEDAI and PZDI adsorbed on the PbI$_2$-terminated surface of FAPbI$_3$ with $I_{Pb}$ antisite defect. **d, e** total DOS calculated for the PbI$_2$-terminated surface of FAPbI$_3$ with $I_{Pb}$ antisite defect passivated with PEDAI and PZDI. Black lines correspond to the total DOS calculated for the unpassivated perovskite surface with the $I_{Pb}$ defect.

On the other hand, the passivation of the $I_{Pb}$ defect with PEDAI does not eliminate the low-lying defect state at 0.1 eV above the Fermi level, as well as introduces the narrow defect state at 1.66 eV, with the edge of the conductivity band shifted to 1.78 eV. Therefore, from the analysis of the electronic structure of the passivated surface with $I_{Pb}$ antisite defect, one can suggest that PZDI passivation should lead to considerably better solar-cell performance. This observation is in line with the calculated defect profile (Fig. 4).

Since PZDI or PEDAI, both contain cations and anions, these could also interact with other charge defects. From the analysis of the Mulliken charges (Figs. S22 and S23), it is found that in PZDI, the charge distribution in the tail is strongly polarized, with both of I in the -NH$_2$I anchor possessing an excess of the negative charge of −0.68|e|, while the NH$_2$ counterpart is positively charged, with the net charge of +0.34|e|. Here, |e| is an elementary charge. On the other hand, in the case of the more stable trans-isomer the PEDAI molecule one of the -NH$_3$I anchor is polarized with the Mulliken charge on I equal to −0.64|e| and charge on the NH$_3$ counterpart of +0.49|e|, while another -NH$_3$I anchor is overall almost neutral with a little polarization. In the case of the cis-isomer of PEDAI both of the -NH$_3$I anchors are overall almost neutral with a little polarization of charges between I and N. This feature explains the difference in the interaction of PEDAI and PZDI molecules with the perovskite surface and their ability to quench the defects. Theoretical analysis of the change in Gibbs free energy upon adsorption demonstrates that PZDI

molecules bind considerably stronger to the surface with $I_{Pb}$ antisite defect in comparison with PEDAI, with the binding energy 1.54 eV per molecule (1.32 eV for PEDAI). Thus, our theoretical analysis corroborates that PZDI passivates the defective surface with a stronger quenching tendency forming the stable film covering the surface.

The observed differences in the distribution tendencies of PEDAI and PZDI in the HP film, as seen in the ToF-SIMS results (Fig. 3h–j), can be correlated with their distinct characteristics such as stark charge difference, their preference for forming a chain (PEDAI/PZDI along-(100)/(010) direction) and surface binding energy (1.32/1.54 eV per molecule for PEDAI/PZDI). These characteristics could play a crucial role in the distribution of PEDAI or PZDI in the HP film. The PEDAI molecule having (100) preferential direction and less binding energy promotes the surface coverage as well as the penetration into the bulk, which explains the distinct variations observed in the ToF-SIMS results. Thus, penetration of PEDAI into the perovskite film tends to reduce conductivity, and therefore, one might expect a decrease in device performance. However, since surface passivation is still effective (Fig. 5d), an improvement in the $V_{OC}$ is achieved. This explains the observed characteristics of low $J_{SC}$, low FF, but high Voc, along with enhanced stability compared to the control device. Therefore, the overarching strategy here is to find a species that stays primarily on the surface and penetrates into the bulk only through defects on grain boundaries, without significantly compromising crystallinity. Having a similar molecular structure to PEDAI and PZDI, one might anticipate a

similar effect. However, we stress that their structural integrity when they are placed at the actual interface matters.

Furthermore, to confirm experimentally the effectiveness of molecular passivation using PZDI, we employed the surface treatment method in perovskite derivatives with varying optical $E_g$. The detailed fabrication method for the devices has been explained in the experimental section. Regarding the wide bandgap HP ($E_g$-1.72 eV, WB-HP; $FA_{0.84}Cs_{0.12}Rb_{0.04}Pb(I_{0.63}Br_{0.37})_3$), we observed that the application of PZDI treatment led to a significant improvement in the *PCE*, increasing it from 17.53 to 19.42% with a notable reduction in the $V_{OC}$ deficit from 0.448 to 0.404 V (Table 2). The *J-V* curves, statistics, and *EQE* spectra are given in Fig. S25 and Table S5. We attribute the reduction in $V_{OC}$ deficit to the mitigation of non-radiative recombination and an enhancement in the quality of the interface achieved through PZDI passivation[39].

Similarly, for the narrow bandgap HP ($E_g$-1.26 eV, NB-HP; $FA_{0.85}MA_{0.1}Cs_{0.05}(Pb_{0.5}Sn_{0.5})I_3$) -based HPSCs, the device results of control and surface-treated NB-HPSCs are given in Fig. S26 and Table S6. The NB-HPSCs with PZDI passivation demonstrated significant improvement in *PCE* of 20.32% with device parameters; $J_{SC}$ -31.46 mAcm$^{-2}$, $V_{OC}$ -0.860 V, and *FF* -0.751. While the control device has a *PCE* of 16.85% ($J_{SC}$ -30.72 mAcm$^{-2}$, $V_{OC}$ -0.785 V, and *FF* -0.730). The device efficiency of NB-HPSCs using Sn/Pb binary HP materials with reduced MA is in the competitive range of another report[73]. The $V_{OC}$ deficit of NB-HPSCs is significantly lowered from 0.478 to 0.405 V (Table 2) which is attributed to attenuation of surface or bulk recombination. This result further confirms the effectiveness of PZDI treatment for NB-HPSCs, consolidating both experimental results and theoretical observations. Importantly, this report corroborates the universality of surface treatment using bifunctional diammonium molecules for the enhancement of device *PCE* and stability by mitigating detrimental defects by modifying the surface and bulk defect chemistry of perovskite film. This observation is parallel to other reports of similar molecular derivatives[45,48–50,74]. While there is still considerable room for improvement, it is noteworthy that, to the best of our knowledge, the performance metrics of these devices fall within the range of the highest reported *PCE* for HPSCs using different perovskite band derivatives.

## Operational stability and monitoring of HPSC degradation

Despite continuous breakthroughs in device efficiency, the stability of HPSCs is still a stumbling block to their competitive reliability in practical applications. To evaluate the device stability, we tracked the device parameters of HPSCs (encapsulated) at the maximum power point tracking (MPPT) conditions under 1 sun irradiation under heat, light, or humidity stress. The device stability data under different aging stress conditions (ISOS-L-2, ISOS-L-3, procedure)[75] were recorded (Fig. 6 and Figs. S27–28 and Tables S7–8). As shown in Fig. 6a, b, the device with PZDI treatment demonstrated superior operational device stability under respective monitoring conditions. Interestingly, despite the ineffectiveness in improving device performance, the PEDAI-treated device showed better device stability compared to the control device suggesting the beneficial effect of surface passivation. At elevated temperature (~60 ± 5 °C; ~35–40% RH), the performance of the control device dropped to ~57.82% of initial *PCE* in 1000 h which

significantly lowered to ~37.80% in 200 h under heat and moisture stress (T = 35 ± 5 °C; RH- 60–65%). Similarly, HPSCs with PZDI treatment retained ~89.48% and 86.74% of original PCE under respective aging conditions. While the PEDAI treated device demonstrated comparatively better device stability than the control device retaining 74.32% and 72.67% of the original *PCE* under respective aging conditions. These data corroborate that the surface treatment with DIM multifunctional molecules significantly improves the device stability under thermal and humidity stress as a consequence of propitious surface chemistry and interfacial surface modulation with strong adsorption energy[31]. This observation indicates that the superior device stability stems from better interface quality and moisture stability in the surface-treated HPSCs.

To consolidate the superior moisture stability data, the water contact angles were taken to study the hydrophobicity of respective HP films (Fig. 6c–e). We noticed a significant drop in water contact angle from 66.42° ($t_0$- 0 s) to 42.36° (t- 1 min) for the control film. The PEDAI-treated HP film shows a higher contact angle of 84.50° ($t_0$- 0 s) which retains at 78.40° after 1 min. Similarly, the PZDI-treated HP film demonstrates a contact angle of 90.60° ($t_0$- 0 s) to 86.20° (t- 1 min). It corroborates that the HP films with surface treatment result in excellent moisture tolerance. The moisture resistivity of the passivated film is attributed to its dense distribution on the film surface (Fig. 3h–j), which agrees with the trend of device stability under higher humidity stress.

Moreover, to contemplate the interfacial deterioration under aging conditions, the capacitance-voltage curves of aged devices were measured. A more pronounced *C-V* hysteresis was seen for the control HPSC (Fig. 6f–h) suggesting a deteriorated interface compared to the HPSCs with surface passivation. This observation substantiates that the control HPSC degrades due to the corrosion of the interfacial junction and increasing dominance of accumulated ions at the interface[76]. A sharp transition of the *M–S* curve in the device with PZDI treatment indicates a smaller depletion layer capacitance ($C_{dl}$) that is attributed to low interfacial defect density. It retains a more stable interfacial junction that stems from intact bulk capacitance ($C_g$)[77,78]. One can see a plateau capacitance ($C_s$) region for $V > V_D$ which is correlated to interfacial charge accumulation and electrode polarization. A suppressed *C-V* hysteresis in PZDI-treated HPSC signifies the suppression of ionic motion or interfacial charge accumulation induced with scan directions[65,79,80]. This result is in line with an earlier report on interfacial degradation analysis[7]. Interestingly, the M-S characteristic features for the aged PEDAI device are not as intact as the PZDI-treated device. These characteristic disparities indicate that alky amine is rather meticulous for surface passivation as supported by theoretical calculations. Thus, this work corroborates that localized electron density in alkyl amine enhances the interfacial adhesion stabilizing the interface and bulk that is benign for device efficiency and operational stability.

In summary, we demonstrated a very effective approach to mitigate the defects in MA-free perovskites using charge-modulated bifunctional molecules with amine terminal. It is found that the diammonium iodide with aryl or alkyl core amine has a remarkable effect on both the efficiency and stability of inverted HPSCs. The PEDAI with a

**Table 2 | Photovoltaic parameters HPSCs (control and target; with PZDI treatment) using perovskite absorber with wide and narrow bandgap (WB, NB)**

| Perovskite system | Device | $E_g^a$ (eV) | $J_{SC}$ (mAcm$^{-2}$) | $V_{OC}$ (V) | FF | PCE (%) | $V_{OC}$ deficit (V) |
|---|---|---|---|---|---|---|---|
| MA-free- WG-HP | Control | 1.722 | 18.80 | 1.274 | 0.732 | 17.53 (16.76 ± 0.69) | 0.448 |
| | Target | 1.725 | 19.32 | 1.321 | 0.761 | 19.42 (18.46 ± 0.42) | 0.404 |
| NG-HP | Control | 1.263 | 31.15 | 0.785 | 0.730 | 17.85 (16.96 ± 0.77) | 0.478 |
| | Target | 1.265 | 31.46 | 0.860 | 0.751 | 20.32 (19.17 ± 0.56) | 0.405 |

The bandgap ($E_g^a$) is estimated from *EQE* analysis. $V_{OC}$ deficit calculated with reference to $E_g^a$. The device statistics are given in parentheses; average values of *PCE* and standard deviation.

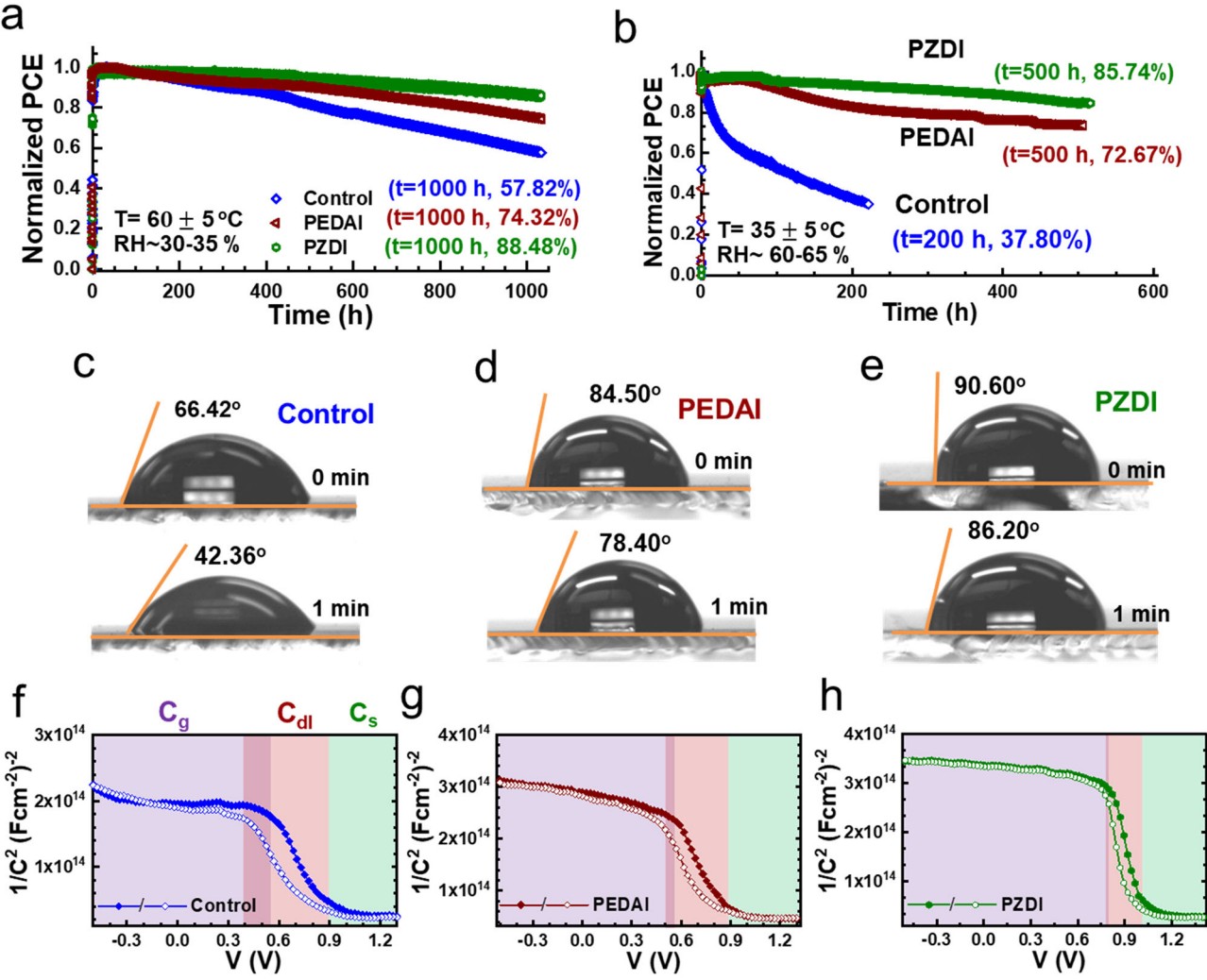

**Fig. 6 | Operational stability of PSCs, hydrophobicity of film, and capacitance response of devices. a**, **b** Device stability monitoring under MPPT conditions: T = 60 ± 5 °C; 30–35% RH (ISOS-L-2, procedure) and T = 35 ± 5 °C; RH- 60–65% (ISOS-L-3, procedure). **c**–**e** Images of the water contact angle on the surface of control, PEDAI, and PZDI perovskite films at different water loading times (initial (0 min) and after 1 min). **f**–**h** M–S plots (open/filled symbols: reverse scan direction) of aged HSPCs (T = 60 ± 5 °C; RH- 30–35%; 1000 h). The color-shaded region represents the characteristic capacitance regime. Overlapped shaded regions represent the lagging of the M-S curve induced by interfacial deterioration.

delocalized lone pair of electrons with an aryl core exerts weak interaction on the surface defects. The PZDI molecules with alkyl core are rather propitious for the improvement in film quality inducing a surface chemical environment for quenching the surface and bulk defect due to stronger interaction with localized electron density. Consequently, the *PCE* of HPSCs with PZDI treatment has been boosted from 19.68% to 23.17% with a large area of ≥ 1 cm² and a reduced $V_{OC}$ deficit of 0.327 V. The molecular distribution of surface-treated film reveals a uniform surface coverage of PZDI with scant diffusion through the bulk layer from ToF-SIMS mapping. Moreover, the PZDI treated-perovskite film exhibited superior operational device stability under heat and humidity stress compared to PEDAI or control devices. This report corroborates that the amine with an alkyl core is crucial for enabling molecular interaction for defect passivation. Hence it paves a new way for designing molecular passivators with charge-regulated molecular bonding for HPSCs and other optoelectronic applications.

## Methods

### Materials and precursor solution
All chemicals were purchased from commercial suppliers as mentioned and unless otherwise specified, they were used as received.

Formamidinium iodide (FAI, GreatCells), 1,4-phenylenediamine dihydriodide (PEDAI, TCI, >98.5%), piperazine dihydriodide (PZDI, TCI, >98.5%). PbI₂ (Wako Chemicals, >98.5%), [2-(3,6-Dimethoxy-9H-carbazol-9-yl) ethyl] phosphonic Acid (MeO-2PACz) (TCI), C₆₀ (TCI, 99%), and Bathocuproine (BCP) (Sigma Aldrich, 99% purity) were purchased and used as received. We used the NiOₓ target (purity >99.9%) from Kojundo Chemical Laboratory Co. Ltd, Japan.

### Fabrication of MA-free HP; FA₀.₈₄Cs₀.₁₂Rb₀.₀₄PbI₃
MA-free halide perovskite: the precursor solution (1.05 M) was prepared by dissolving FAI (0.84 M), CsI (0.12 M), RbI (0.04 M), PbI₂ (1 M), and 5-AVAI (1 mM) in the mixture of dimethylformamide (DMF), and dimethyl sulfoxide (DMSO) (4:1) solvent for 2 h at 60 °C temperature. For film deposition, the precursor was spin-coated at 1000 rpm for 10 s (ramping slope 2 s) and 5000 rpm for 40 s followed by dripping 800 μl of CB at the 35th second of the second step. Then to promote the crystallization, those as-grown films were simply placed on a hot plate at 60 °C for 1 min with further annealing at 100 °C for 45 min For the surface treatment strategy, the PEDAI or PZDI precursor solutions of various concentrations (0.5, 1, 2 mg/ml) were prepared by dissolving in isopropyl alcohol (IPA) at 60 °C for 2 h. For surface passivation, the PEDAI or PZDI solutions were spin-coated onto the HP film at

5000 rpm for 40 s (ramping slope 3 s) and annealed at 100 °C for 5 min.

All the solutions were filtered using 0.2 μm syringe filters just before the deposition to avoid the risk of unwanted particles in the precursor solution.

## Perovskite solar cell Fabrication

Solar cell devices were fabricated on pre-cleaned patterned indium tin oxide (ITO) coated glass substrates (15 Ω square$^{-1}$). The ITO substrates (4.5 cm × 3.5 cm) were pre-cleaned in an ultrasonic bath with detergent, pure water, and 2-propanol, followed by an ultraviolet-ozone treatment for 5 min to remove the organic residuals. The $NiO_x$ (~20 nm) film was deposited by sputtering as mentioned in our earlier reports[7]. In brief, the pre-cleaned ITO substrates were loaded in the deposition chamber and evacuated until <2×10$^{-3}$ Pa then pure argon gas was introduced at the rate of 20 sccm. The $NiO_x$ deposition was carried out in an argon gas pressure of 3.5 Pa and rf power supply of 150 W for 7 min at room temperature. Then substrates were transferred into a nitrogen-filled glove box (<1.0 ppm $O_2$ and $H_2O$) and the rest of the steps were carried out inside the glove box. The sputtered $NiO_x$ thin film was treated with MeO-2PACz (0.1 weight% in ethanol) by spin coating at 5000 rpm for 50 s and subsequently dried at 100 °C for 10 min on a hot plate. For surface treatment, the PEDAI or PZDI precursors dissolved in IPA were spin-coated onto the HP film at 5000 rpm for 40 s (ramping slope 3sec) and annealed at 100 °C for 5 min The $C_{60}$ (26-28 nm at 0.1Å/s) layer was deposited by thermal evaporation as an electron transport layer (ETL). After that, the electron selective layer (ESL), BCP (5–6 nm, at 0.01 Å/s) was deposited by evaporation at a base pressure of ~2×10$^{-4}$ Pa. Then, to complete the device structure, samples were then transferred into the evaporation chamber connected to the glove box for metal contact deposition. Finally, 140 nm of Ag was thermally evaporated at a pressure <10$^{-4}$ Pa. Four devices in ITO substrate (4.5 cm × 3.5 cm) with an active device area of ~1.26 × 1.26 cm$^2$ were sealed using UV-curable resins before the subsequent measurements in ambient conditions.

## Fabrication of other perovskites system devices

### Fabrication of wide-bandgap (WB) halide perovskite (WB-HPSCs)

Preparation of MA-free WB-HP; FA$_{0.84}$Cs$_{0.12}$Rb$_{0.04}$Pb(I$_{0.63}$Br$_{0.37}$)$_3$. Wide bandgap halide perovskite (WB-HP): the precursor solution (1 M) was prepared by dissolving FAI (0.84 M), CsI (0.12 M), RbI (0.04 M), PbI$_2$ (0.45 M), PbBr$_2$ (0.55 M), and 5-AVAI (1 mM) in 1 ml of mixture of DMF and DMSO (4:1) solvent for 2 h at room temperature. Note that we adopted cations composition as in regular MA-free HPSCs while the halide composition was taken with the reference of MA-free WB-HP reported by the Snaith group[81] and our previous report[71]. For film deposition, the precursor was spin-coated at 1000 rpm for 10 s (ramping slope 2 s) and 5000 rpm for 40 s followed by dripping 800 μl of CB at the 42th second of the second step. Then to promote the crystallization, those as-grown films were simply placed on a hot plate at 60 °C for 2 min and at 100 °C for 45 min The surface passivated devices were prepared by spin coating PZDI same as regular bandgap HP devices. the ETL layer; $C_{60}$-fused *N*-methylpyrrolidine-*meta*-dodecyl phenyl (C$_{60}$MC$_{12}$) was used as reported in our earlier report for WB-HHPSC[71].

Finally, we fabricated WB-HP-based HPSCs of inverted configurations; ITO/NiO$_x$/MeO-2PACz/WB-HP/C$_{60}$MC$_{12}$/BCP/Ag.

### Fabrication of narrow bandgap (NB) halide perovskite (NB-HPSCs)

Preparation of NW-HP; FA$_{0.85}$MA$_{0.1}$Cs$_{0.05}$(Pb$_{0.5}$, Sn$_{0.5}$)I$_3$. Narrow bandgap halide perovskite (NB-HP): the precursor solution (1.4 M) was prepared by dissolving FAI (0.85 M), MAI (0.1 M), CsI (0.05 M), PbI$_2$ (0.5 M), SnI$_2$ (0.5 M), SnF$_2$ (0.05 M), and MASCN (0.05 M) in the mixture of DMF and DMSO (4:1) solvent for 1 h at 50 °C temperature. Note that the NB-HP; Cs$_{0.05}$MA$_{0.1}$FA$_{0.85}$(Pb$_{0.5}$, Sn$_{0.5}$)I$_3$; composition was inspired from a two-step process report by Yanfa Yan and co-workers[73].

We have used a single precursor solution for the preparation of NB-HP film. For film deposition, the precursor was spin-coated at 5000 rpm for 50 s (ramping slope 3 s) followed by dripping 150 μl of CB at the 20th second of the second step. Then, as deposited films were placed on a hot plate at 60 °C for 2 min and at 100 °C for 10 min For surface treatment with PZDI, the PZDI solution in IPA was spin-coated onto the NW-HP film at 5000 rpm for 40 s (ramping slope 3 s) and annealed at 100 °C for 5 min.

Finally, we completed NB-HP-based HPSCs of inverted configurations; ITO/PEDOT:PSS/NB-HP/C$_{60}$/BCP/Ag.

## Synthesis of PZDI-based single crystals

A single crystal was synthesized by the antisolvent vapor-assisted crystallization method reported by Bakr and co-workers[82]. Two sets of precursors (5 ml) were prepared by dissolving (1) PbI$_2$ (1 M) + PZDI (1 M) (1:1) and (2) PbI$_2$ (0.5 M) + PZDI (1 M) (1:2) were dissolved in DMSO and stirred overnight. The precursor solution was filtered and transferred into 10 mL vials. The vial with an opened lid was then placed into a sealed bottle filled with 3 mL tetrahydrofuran (THF) as an antisolvent. The single crystals were grown along with the slow diffusion of the vapor of the anti-solvent THF into the precursor solution. Finally, single crystals were obtained after keeping the precursor without disturbing it for 72 h. These single crystals with DMSO complex were analyzed by measuring single crystal XRD.

## Materials and device characterizations

In NIMS Battery Research Platform facilities, X-ray diffraction (XRD) patterns of fabricated MA-free-HP films were collected using an advanced X-ray diffractometer (Rigaku SmartLab, CuK$_α$ radiation, λ = 1.54050 Å). The stresses were measured according to the 2θ–sin2 φ method using Cu-Kα radiation in a RIGAKU SmartLab X-ray diffractometer and beam/parallel slit analyzer (PB/PSA) optics. The diffracted rays were measured at different angles (tilt angle, ψ = 10°–60°) by fixing a diffraction plane of (012) for e perovskite film. Grazing Incidence X-ray Diffraction (GIXRD) analysis was measured with the Rigaku SmartLab diffractometer (Rigaku SmartLab, CuK$_α$ radiation, λ = 1.54050 Å) with grazing incident angle (ω– 0.05°, 0.1°, 0.2°,……3°.). The single crystal XRD data was collected by XtaLAB Synergy-DW (Mo) and XtaLAB miniII (X-ray source-Mo Kα (λ = 0.71073 Å, 50 kV–24 mA, λ = 1.54056 Å) in Rigaku R & D facilities in Japan. X-ray photoelectron spectroscopy (XPS) spectra were obtained using a Versa Probe II (ULVAC-PHI, Japan). Perovskite film samples for XPS measurements were prepared in an N$_2$-filled glove box and transferred to the XHPSChamber through an N$_2$-filled transfer vessel in order to avoid oxygen contamination. XPS with a nonmonochromatic source was measured (Al Kα; 1486.6 eV, spot size 10–300 μm) at a pass energy of 187.85 eV (1.5 eV step size) for the survey scan and pass energy 46.95 eV (0.1 eV step size) for the fine scan with spot size 100 μm. The XPS spectra were calibrated with the binding energy of 284.8 eV for C1s.

In NIMS Namiki foundry research facilities and category-3 shared facilities, the morphology of films and cross-sectional images were taken by a high-resolution scanning electron microscope (SEM) at 5 kV accelerating voltage (Hitachi, S-4800). The photoluminescence (PL) spectra were collected using the micro-PL spectrometer (HORIBA, LabRamHR-PL NF(UV-NIR)) ~532 nm laser diode (10 mWcm$^{-2}$) as an excitation source. The carrier lifetimes were measured with a fluorescence lifetime spectrometer (Quantaurus-τ from Hamamatsu-Photonics K.K., C11367) equipped with ~405 nm laser diode (typical peak power of 400 mW) at 200 kHz repetition rate. The absorption spectra films were measured using the UV-Vis-NIR spectrometer (UV-2600i, Shimadzu). The absorption spectra and photoluminescence (PL) spectra of various films were measured using the UV-Vis-NIR spectrometer (UV-2600i, Shimadzu). The band structure of the film was measured using Ultraviolet photoelectron spectroscopy (UPS, Thermo Fisher Scientific, Inc.) with a He I line (21.22 eV) from a helium discharge lamp.

The samples for transmission electron microscopy (TEM) were prepared by using a focused ion beam (FIB) technique (JEOL, JIB-4501) inside a glove box. Before the sample preparation with the FIB, we deposited a thin layer of osmium and carbon on the top of the sample (Ag layer peeled off) to protect it from damage during milling. Sample extraction was performed with an FIB accelerating voltage of 30 kV and a current of 800 pA. Once the lamella was extracted and welded on the Mo grid, it was thinned to electron transparency at 10 kV and 10 pA. The SEM accelerating voltage was kept at 5 kV for the entire process. The TEM samples were about 50–60 nm thick. The finished TEM was immediately transferred for TEM analysis to limit overall exposure to air to <2 min STEM/EDX was carried out at room temperature (~300 K) using an analytical TEM (JEOL JEM-ARM200F for HR, a 200 kV acceleration voltage) equipped with a cold-field emission gun and a JEOL EDX detector. To minimize the damage from the electron beam, we limited the exposure time to within 5 s in every TEM observation.

Time-of-flight secondary ion mass spectrometry (ToF-SIMS) measurements were carried out using a ToF.SIMS 5 (ION-TOF GmbH) instrument equipped with a 60 keV and pA current $Bi^{2+}$ beam for analysis and a 10 keV and nA current Ar gas cluster ion beam (Ar-GCIB) for sputtering in non-interlaced mode to have minimal interfacial mixing. The sputtered area was $700 \times 700\ \mu m^2$ and the analysis area was $100 \times 100\ \mu m^2$.

The current density–voltage (J–V) curves were measured at the scan rate of 0.05V/s under 1 sun with an AM1.5G spectral filter (100 mW/cm²) coupled with an MPPT system (Systemhouse Sunrise Corp.). Each device's efficiency was confirmed under MPPT for 2 min of tracking. The light intensity was calibrated by a silicon (Si) diode (BS-520BK). For the stability test, the encapsulated devices were measured at MPPT conditions. The devices were kept under 1-Sun intensity under MPPT conditions (elevated temperature, 60 ± 5 °C, relative humidity, 30–35% RH and ~35± 5 °C; 60–65% RH, RT) during device stability monitoring (Bunkoukeiki Corp. BIR-50 solar cell light resistance test system, incubator type 50 × 50 mm irradiation, Systemhouse Sunrise Corp.). The external quantum efficiency (EQE) spectra were obtained using a spectrometer (SM-250IQE, Bunkokeiki, Japan).

The transient photovoltage was measured using a commercial PAIOS system (PAIOS V.4.3). A pulse intensity was used to induce a spike in photovoltage. The capacitance spectra (C-f) were taken from PAIOS v. 4.3 software, which scans from 20 Hz to 2 MHz at 30 mV AC in the dark at a bias voltage of 0 V. The $C-V$ measurements were taken at 10 kHz with a voltage amplitude of 30 mV AC in the dark. The scan frequency is determined from the plateau region (corresponds to $C_g$) of the capacitance-frequency spectra $C-f$ scan at zero bias. The thermal capacitance spectra (C-f-T) were measured using an LCR meter (IM3536, Hioki), with a voltage amplitude of 30 mV under dark conditions in the temperature range of 253–353 K. The temperature was varied by using the controlled chamber (SU-221) (±0.1 K).

### Solar cell certification
Device certification was obtained from The National Institute of Advanced Industrial Science and Technology (AIST), Japan. It is registered as ISO / IEC 17025 accreditation laboratory (IAJapan ASNITE 0021 Calibration) according to international mutual recognition arrangements (MRA) for ILAC and APAC.

### Theoretical and computational methods
Density functional theory (DFT) calculations have been performed using the fully constrained meta-generalized-gradient approximation (meta-GGA) SCAN[83] for the exchange–correlation functional and the projector-augmented wave (PAW) formalism as implemented in the Vienna ab initio simulation package (VASP)[84,85]. This approximation has previously demonstrated a remarkable accuracy for the description of lattice constants and weak interactions in a large diversity of bonded molecules and materials[83,86], which is especially important for the investigation of the complicated process of molecular adsorption and passivation of perovskite surfaces. The pseudo-cubic structure of the model $FAPbI_3$ bulk was optimized using the regular Γ-centered $6 \times 6 \times 6$ k-point mesh for Brillouin zone sampling. The energy cutoff of 500 eV for the plane wave basis set and the convergence criterion of $10^{-6}$ eV for the self-consistent loop were employed. The optimized lattice parameters of the pseudo-cubic $FAPbI_3$, a = 6.4782 Å, b = 6.3080 Å, c = 6.4012 Å are in good agreement with an experimentally observed cubic unit cell of dimension ~6.36 Å[87]. The optimized lattice of bulk $FAPbI_3$ was used to construct 2x2 slab of $PbI_2$-terminated surface (001) with five $PbI_2$ layers and a vacuum region of ~25 Å. Two top $PbI_2$-FAI bilayers were fully relaxed, while the atoms in the bottom layers were frozen. For the Brillouin zone sampling of the slab, the Γ-centered $3 \times 3 \times 1$ and $6 \times 6 \times 1$ k-point meshes have been used for structural relaxations and DOS calculations, respectively.

The adsorbed PEDAI and PZDI molecules have been optimized on the perovskite surface, where all atoms in the molecules and two top $PbI_2$-FAI bilayers were fully relaxed until forces were <0.01 eV $Å^{-1}$. The binding energy of the molecule to the surface was evaluated from the change in Gibbs free energy as $E_b = -(\Delta E_{el} - T\Delta S + \Delta E_{ZPE})$. Here $\Delta E_{el} = E_{passivatedsurface} - (E_{surface} + E_{mol})$, where $E_{passivatedsurface}$ is the total energy of the passivated surface, while $E_{surface}$ and $E_{mol}$ are energies of the free surface and the isolated PEDAI or PZDI molecule, as follows from DFT calculations. T is the temperature of the system taken equal to 293 K, $\Delta S$ is the change of entropy upon molecular adsorption, and $\Delta E_{ZPE}$ is the change in zero-point energy. To estimate the change in entropy, we have used the ideal gas approximation for the free molecules as implemented in Gaussian 09 and considered S = 0 for the adsorbed molecules, because they are immobilized on the surface, losing the translational and rotational degrees of freedom.

### Reporting summary
Further information on research design is available in the Nature Research Reporting Summary linked to this article.

## Data availability
The data that support the findings of this study are available from the corresponding authors upon request.

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

## Acknowledgements

The authors are grateful to Yamaguchi Kazuo-San (XPS), Nariaki Sato-San (ToF-SIMS), and Keisuke Shinoda-san/Makiko Oshida-san (TEM) from the NIMS battery research platform for the technical support for respective measurement and analysis. We are thankful to Rigaku R & D facilities in Japan for carrying out single crystal XRD measurements. Calculations were performed using computational resources of the Institute for Solid State Physics, the University of Tokyo, Japan; and the Research Center for Computational Science, Okazaki, Japan (Project: 23-IMS-C016).

## Author contributions

D.B.K. conceived the idea, designed, and performed the device fabrication work. Y.S. and M.Y. optimized the hole transport layer and validated device results. M.Y. and D.B.K. performed UPS measurement and data analysis. D.B.K. with technical staff collected STEM, ToF-SIMS, and XPS data and performed data analysis. H.O. with D.B.K. performed and analyzed XRD measurements. D.B.K. carried out device characterizations and analysis. A.L. and T.T. provided theoretical support. K.M. supervised the analysis of material characterizations and device analysis. D.B.K. wrote the original manuscript. All authors discussed the results and reviewed the manuscript.

## Competing interests

The authors declare no competing interests.

## Additional information

**Supplementary information** The online version contains
supplementary material available at

Dhruba B. Khadka, Yasuhiro Shirai or Andrey Lyalin.

**Peer review information** *Nature Communications* thanks the anon-
ymous reviewer(s) for their contribution to the peer review of this work. A
peer review file is available.

