## [Peer Review File · Nature Communications]

Defect Passivation in Methylammonium/Bromine Free
Inverted Perovskite Solar Cells Using Charge-Modulated
Molecular BondingReviewers' comments:

Reviewer #1 (Remarks to the Author):

The authors reported a piperazine diiodide as a surface/interface modifier to passivate the defects on perovskite for improving the power conversion efficiency (PCE) and stability of perovskite solar cells, and an outstanding PCE up to 23.17% has been achieved. However, I can not suggest to publish it in Nat. Comm. at least in this stage due to the following reasons.

1. Piperazine diiodide has been applied in perovskite solar cells (<https://doi.org/10.1002/sml.202208260>; <https://doi.org/10.1039/C9TC03576A>) as passivator or forming 2D structure to improve perovskite stability. It seems no new effects has been found in this paper.
2. A variety of certified PCEs of over 23% for 1 cm² PSCs (<https://doi.org/10.1002/pip.3726>; <https://www.science.org/doi/10.1126/science.abn3148>; <https://doi.org/10.1002/adma.202205027>) was overlooked by the authors.
3. The initial PCEs (Fig. 6) should be reported for the stability test.

Reviewer #2 (Remarks to the Author):

This paper reports the effects of the surface passivation of Pb perovskite with diammonium iodide (PEDAI and piperazine dihydriodide PZDI). Based on this surface passivation for p-i-n device, the performance and stability are improved, and the present approach seems to be very reasonable. However, the very similar approach using diammonium halide on the top surface of p-i-n type device have already reported in details by the other group (EDAI2: S. Hu, et al. Energy Environ. Sci. 2022, 15, 2096 for Sb-Pb mixed p-i-n system, ACS Appl. Mater. Interfaces, 2022, 14, 56290 for both Pb and Sn systems, and piperazine and its derivatives: Adv. Mater. 2023, 35, 2208320, etc.). The authors do not mention these works in the paper at all, unfortunately. The authors should note these pioneering research results in the introduction section adequately, and make clear what is the new finding clear on the authors' MA-free Pb-system over the previous reports. Therefore, on the current version, I cannot support this publication. I might be able to support this publication if the authors could adequately and clearly show any new findings.

- 1) As mentioned above, firstly, the authors should introduce the previous work adequately, especially very relating work on diammonium halide. The applicability of diammonium halide, EDAI2, as dipole strategy for both Pb and Sn-based perovskite solar cells have already been shown in ACS Appl. Mater. Interfaces, 2022, 14, 56290. In addition, the molecular structural effects using the piperazine and its

derivatives on the stability have been reported in details in Adv. Mater. 2023, 35, 2208320. The present approach and results in this paper supports the rationality and utility of these previous studies.

2) Although the authors discuss the surface structure formed by PEDAI and PZDDI passivation by assuming the formation of 2D structure using DFT modeling and the results of film XRD. The accommodation of ammonium alkyl group at the A-site on the 2D perovskite should highly depends on the bulkiness. In the discussion of the structural differences of surfactants, the authors could gain the clear information about the surface structure by using single crystal X-ray structural analysis for the model 2D or 1D compound prepared from the surfactant and simple APbI₃ (see the works by S. Hu et al.). Since the current observation and discussion is just based on the ambiguous structure as "2D perovskite" just based on DFT modeling and less information from the film XRD, I strongly recommend the confirmation the surface structure using the single crystal XRD.

3) Regarding the post surface treatment, the condition (concentration) and method of the solution should be critical in terms of reproducibility. The use of IPA solution could dissolve the surface of 3D-perovskite layer as well as PbI₂ and work to form 2D-like layer at the surface. The authors should give the details more clearly.

Reviewer #3 (Remarks to the Author):

In this manuscript, the authors improved the performance of inverted PSCs by top surface passivation employing piperazine dihydriodide which possesses an alkyl ring structure. The molecular possess two –NH groups, which can passivate the surface defect, meanwhile promote the carrier extraction through surface chemistry and band energy modification. The authors extensively studied the working mechanism from the viewpoint of structural, optical and electrical effect, and demonstrated the universality in wide bandgap and narrow bandgap PSCs.

However, plenty of molecules containing ammonia group have been employed and depicted effectiveness for surface passivation, hence the manuscript exhibited less novelty especially considering the Nature Communications is a multidisciplinary Journal. Current manuscript is more suitable to be published on Journals focusing on energy or materials.

Several suggestions also listed below:

1) The champion PCE should be updated.

2) PEDAI is easy for 2D material formation, while PZDI prefers to cover on the grain surface. In addition, PEDAI is easier to penetrate into perovskite, while which has similar size with PZDI. The detailed mechanism should be explained.

3) The better EQE response in $450 < \lambda < 330$ nm is attributed to betterment in interface quality of PZDI-treated HPSC, which is not convinced. As the photon in this part are mainly absorbed around the bottom of active layer.

Research Article- NCOMMS-23-24660A-Z

Title: Defect Passivation in Methylammonium/Bromine Free Inverted Perovskite Solar Cells Using Charge-Modulated Molecular Bonding

Reviewers' comments:

Reviewer #1 (Remarks to the Author):

The authors reported a piperazine diiodide as a surface/interface modifier to passivate the defects on perovskite for improving the power conversion efficiency (PCE) and stability of perovskite solar cells, and an outstanding PCE up to 23.17% has been achieved. However, I cannot suggest publishing it in Nat. Comm. at least in this stage due to the following reasons.

We thank the reviewer for the positive comments and for providing us with constructive suggestions.

We have revised the contents and referred to noted publications, with the aim to improve the manuscript based on the suggestions.

The detailed responses to each point are shown below.

Comments #R1-1

1. Piperazine diiodide has been applied in perovskite solar cells (<https://doi.org/10.1002/sml.202208260>; <https://doi.org/10.1039/C9TC03576A>) as passivator or forming 2D structure to improve perovskite stability. It seems no new effects have been found in this paper.

We thank the reviewer for sharing interesting reports.

We would like to note that this work is different from these reports.

In an above-mentioned report (<https://doi.org/10.1002/sml.202208260>), they used piperazine dihydriodide in halide perovskite (A site (FA, MA, Cs) solar cells on HTL surface (not on perovskite surface). They have mainly discussed strain analysis with HTL interface passivation.

In another report (<https://doi.org/10.1039/C9TC03576A> and device with carbon electrode with comparatively low device efficiency), they used as an additive in the bulk perovskite (all inorganic).

In our report, we investigated the detailed insights of molecular passivation with PEDAI and PZDI comparing their functionalities.

We believe that our report unraveled a new effect of charge distribution in molecular passivation; PEDAI and PZDI in perovskite solar cells.

Despite similarities in chemical functionality, we found a distinct effect of PEDAI and PZDI in the film growth, materials distributions, and device properties. These results are thoroughly explored coupling a set of experimental data and theoretical calculations.

Comments #R1-2

2. A variety of certified PCEs of over 23% for 1 cm² PSCs (<https://doi.org/10.1002/pip.3726>; <https://www.science.org/doi/10.1126/science.abn3148>; <https://doi.org/10.1002/adma.202205027>) was overlooked by the authors.

We are thankful to the reviewer for sharing interesting large area (1cm²) device reports. Although we missed a few interesting reports as noted by the reviewer, we had listed reports with competitive PCE over 23%.

Considering reviewer's comment, we have added above mentioned reports (<https://www.science.org/doi/10.1126/science.abn3148>; <https://doi.org/10.1002/adma.202205027>) in summarized **Table S3**.

We also would like to note that most of the reports used perovskite containing MA and mixed halide and regular device structure.

However, we have used MA-free and mono halide (Iodine) perovskite and inverted structure for the fabrication of HPSCs.

As listed in **Table S3**, our device report is competitive considering MA-free perovskite and p-i-n device structure.

Comments #R1-3

3. The initial PCEs (Fig. 6) should be reported for the stability test.

We thank the reviewer for the constructive comment.

We included the initial device efficiency of the device used for stability testing in the supporting information and revised the related content. We also have included a consensus statement for stability assessment (**ISOS procedures**, Khenkin, M. V. et al. Consensus statement for stability assessment and reporting for perovskite photovoltaics based on ISOS procedures. *Nat Energy* 5, 35–49 (2020)) in the revised manuscript.

Please see **Figures S27 and S28** and summarized **Tables S6 and S7**.

Figure R1. (Figure S27, S28) Stability of the control, PEDAI, and PZDI passivated HPSCs (corresponding to Figure 6); stability monitoring a) operational tracking under MPPT conditions: T=60 ± 5 °C; 30–35% RH (ISOS-L-2, procedure) and b) T=35 ± 5 °C; RH~ 60–65% (ISOS-L-3, procedure).

Reviewer #2 (Remarks to the Author):

This paper reports the effects of the surface passivation of Pb perovskite with diammonium iodide (PEDAI and piperazine dihydriodide PZDI). Based on this surface passivation for the p-i-n device, the performance and stability are improved, and the present approach seems to be very reasonable. However, a very similar approach using diammonium halide on the top surface of a p-i-n type device has already been reported in detail by the other group (EDAI₂: S. Hu, et al. Energy Environ. Sci. 2022, 15, 2096 for Sb-Pb mixed p-i-n system, ACS Appl. Mater. Interfaces, 2022, 14, 56290 for both Pb and Sn systems, and piperazine and its derivatives: Adv. Mater. 2023, 35, 2208320, etc.). The authors do not mention these works in the paper at all, unfortunately. The authors should note these pioneering research results in the introduction section adequately and make clear what is the new finding clear on the authors' MA-free Pb-system over the previous reports. Therefore, on the current version, I cannot support this publication. I might be able to support this publication if the authors could adequately and clearly show any new findings.

We thank the reviewer for the positive and constructive comments.

We also thank the reviewer for sharing startling reports by the Wakamiya group. We apologize for missing these reports in our discussion. We have included these reports in the introduction section and discussion in the main text in the relevant context.

Considering the reviewer's comments, we have revised the content highlighting new insights. We hope the reviewer will be positive for the consideration of the revised manuscript.

The detailed responses to each comment are shown below.

Comments #R2-1

1) As mentioned above, firstly, the authors should introduce the previous work adequately, especially

very relating work on diammonium halide. The applicability of diammonium halide, EDAl₂, as dipole strategy for both Pb and Sn-based perovskite solar cells, have already been shown in *ACS Appl. Mater. Interfaces*, 2022, 14, 56290. In addition, the molecular structural effects using the piperazine and its derivatives on the stability have been reported in detail in *Adv. Mater.* 2023, 35, 2208320. The present approach and results in this paper supports the rationality and utility of these previous studies.

We thank the reviewer for their suggestion on including reported work related to diammonium halide.

We agreed that the diammonium halide derivatives have been used by the Wakamiya group in various perovskite systems as listed in the comment.

Indeed, we were inspired by diammonium halide functional molecules following the previous reports listed below. It is interesting to note the importance of piperazine derivatives in perovskite systems (*Advanced Materials* 35, 2208320 (2023)).

We noted these reports in the introduction and results and discussion section in the revised manuscript.

On page#2 +

The function additives also remove molecular iodine existing in perovskites and quench the iodine and the Pb-related deep trap sites.^{29,30} Importantly, the introduction of molecular functional derivatives of diammonium halide into alkyl or aryl core, serving as additives or passivators, has also exhibited noteworthy enhancements in both the PCE and stability of HPSCs using various perovskite systems.^{31–35} Despite lots of additives being wielded for optimization of perovskite growth and defect passivation, the influence of bonding interaction between additives to perovskite is rarely explored.

31. Hu, S. *et al.* Optimized carrier extraction at interfaces for 23.6% efficient tin–lead perovskite solar cells. *Energy Environ Sci* **15**, 2096–2107 (2022).
32. Song, Q. *et al.* Bridging the Buried Interface with Piperazine Dihydriodide Layer for High Performance Inverted Solar Cells. *Small* **19**, (2023).
33. Wang, H. *et al.* Skillfully deflecting the question: a small amount of piperazine-1,4-dium iodide radically enhances the thermal stability of CsPbI₃ perovskite. *J Mater Chem C*, **7**, 11757, (2019).
34. Hu, S. *et al.* Synergistic Surface Modification of Tin–Lead Perovskite Solar Cells. *Advanced Materials* **35**, 2208320 (2023).
35. Hou, M. *et al.* Aryl Diammonium Iodide Passivation for Efficient and Stable Hybrid Organ - Inorganic Perovskite Solar Cells. *Adv Funct Mater* **30**, 2002366 (2020).

Comments #R2-2

2) Although the authors discuss the surface structure formed by PEDAI and PZDDI passivation by assuming the formation of 2D structure using DFT modeling and the results of film XRD. The accommodation of ammonium alkyl group at the A-site on the 2D perovskite should highly depend on the bulkiness. In the discussion of the structural differences of surfactants, the authors could gain the clear information about the surface structure by using single crystal X-ray structural analysis for the model 2D or 1D compound prepared from the surfactant and simple APbI₃ (see the works by S. Hu et al.). Since the current observation and discussion is just based on the ambiguous structure as "2D perovskite" just based on DFT modeling and less information from the film XRD, I strongly recommend the confirmation the surface structure using the single crystal XRD.

We appreciate the reviewer's valuable suggestion very much regarding the surface structure formed by PEDAI and PZDDI passivation.

However, we are a little perplexed because we are not assuming the 2D structure. We would like to note that 2D perovskite was experimentally observed and supported by DFT calculation. We have confirmed 2D perovskite crystallite from XRD results of surface passivated HP films and HP films prepared by mixing PEDAI or PZDI molecules in perovskite precursor as shown in Fig. R2 (Figure S3). One can see the flex feature on the SEM image of perovskite film with mixed PEDAI (Figure R2/a3) while that for the HP film with mixed PZDI has fewer distinct features (Figure R2/b3). The SEM and XRD observations are also supported by PL spectra (Figure R2/c) We have confirmed it by HR-TEM measurements as shown in Figure S15. A part of interfacial HR-TEM is shown in Figure R3

We have included a related discussion in the revised manuscript. We would like to note that the formation of 2D perovskite with PEDAI and PZDI under surface passivation has a completely different story than the synthesis of a single crystal using PEDAI and PZDI.

Figure R2. Photo of mixed HP precursor [a₁) PEDAI or b₁) PZDI and HaP/mixed precursor]. a₂, b₂) XRD patterns of PbI₂ film, HP film surface treated with PEDAI or PZDI (dissolved 2 mg/ml in IPA), HP film with mixed PEDAI or PZDI, and power crystal prepared by mixing PbI₂ and PEDAI or PZDI

in 1:2 ratio. Here, # -2D phase with PEDAI or PZDI, - PbI₂ peak, δ - non-photoactive perovskite phase, α - photoactive perovskite phase. a₃, b₃) SEM images of HaP film (mixed precursor: PEDAI or PZDI/perovskite-mixed precursor. c) PL spectra of the HaP film prepared using mixed precursor. The shoulder response marked with # in PL spectra stems from the 2D phase formed with PEDAI or PZDI. Note that mixed precursor was prepared by mixing 0.5 M-PEDAI or PZDI + 0.5 M PbI₂ and 1M of control precursor (FA_{0.84}Rb_{0.04}Cs_{0.12}PbI₃).

Figure R3. Cross-sectional HR-TEM image of C₆₀/HaP top surface (a,b,c) with 2D phase interface on the surface of 3D-HP with diffusion through grain boundaries to some extent. (Figure S15)

Moreover, the material, that the reviewer suggested paper (Hu et al.) studies, belongs to the class forming Dion-Jacobson series (JACS 2018, 140, 3775), with documented procedure for single crystal growth. On the other hand, we are not aware of any reports of single crystal growth with PEDAI or PZDI. Developing the protocol for crystal growth is quite an endeavor in itself.

As per the reviewer's recommendations, we proceed with the single crystal synthesis with PZDI molecules. Single crystals were grown by mixing 2:1 and 1:1 molecular ratios of PZDI and PbI₂ adopting the antisolvent vapor-assisted crystallization method as described in supporting information (Figure S4). We obtained single crystal of composition (PZDI)₂(PbI₄)₂ and (PZDI)₃Pb₂I₇ with DMSO complex confirmed by single crystal XRD analysis. Figure R4 (supporting information, Figure S5, Table S1) shows the simulated XRD results and corresponding crystal unit and packing fraction. Although the growth condition is completely different from film preparation, the XRD data of single crystal was found to be close to the synthesized single crystal data. This confirmed the formation of small crystallites of various compositions of PZDI and PbI₂ when the surface of HP film is treated with PZDI molecules or mixed in HP precursor.

We have included this discussion in the revised manuscript.

Figure R4. Analysis of single crystal obtained by adopting the method as depicted in Figure 4. (a₁) simulated PXRD result of crystal (insets are optical image, dimension, and composition: $(\text{PZDI})_2(\text{PbI}_4)_2 \cdot 6\text{DMSO}$) grown in precursor solution (mixing PZDI and PbI_2 in 2:1 molar ratio), (a₂) simulated crystal unit, (a₃) Packing diagram. (b₁-b₃) corresponding results and properties of crystal sample of $(\text{PZDI})_3\text{Pb}_2\text{I}_7 \cdot 6\text{DMSO}$ grown in precursor solution (mixing PZDI and PbI_2 in a 1:1 molar ratio).

We have revised the related content on page#5.

To gain deeper insights into the formation of the 2D phase, we prepared HP film by mixing DIM in the perovskite precursor solution, as depicted in supporting information (Figure S3a₁, b₁) and powder crystal by mixing (PbI_2 and PEDAI or PEDAI). XRD results revealed that the HP films with DIM mixing and powder crystal grow without any remanence of PbI_2 peak suggesting a strong tendency for the formation of a new crystal phase. Besides that, XRD patterns (Figure S3a₂, b₂) demonstrated a more intensified XRD peak at $2\theta < 10^\circ$ for the HP film prepared with mixed DIM compared to the one subjected to the surface treatment alone. Importantly, the HP film with mixed PEDAI was found to be grown with much higher XRD peak intensity at $\sim 4.8^\circ$ indicating a more preference towards 2D phase formation. This observation was further supported by SEM images as displayed in supporting information (Figure S3a₃, b₃) which depicted nano-sheet-like features in the HP mixed containing PEDAI, underscoring the presence of PEDAI-based 2D phase. This observation is consistent with the small crystallite observed in the PEDAI passivated film. Interestingly, in contrast to the PEDAI-mixed scenario, the HP film with mixed PZDI exhibited growth as an overlayer on crystal grains or at grain boundaries, rather than forming a flake structure that closely resembled that of the passivated film. To confirm the single crystal

structure of the 2D phase with PZDI in HP film, single crystals were grown by mixing 2:1 and 1:1 molecular ratios of PZDI and PbI_2 using the reported antisolvent vapor-assisted crystallization method as described in supporting information (Figure S4). We obtained single crystal of composition $(\text{PZDI})_2(\text{PbI}_4)_2$ and $(\text{PZDI})_3\text{Pb}_2\text{I}_7$ with DMSO complex confirmed by single crystal XRD analysis. The supporting data (Figure S5) show the simulated XRD results and corresponding crystal unit and packing fraction. Although the growth condition is completely different from film preparation, the XRD data of single crystal was found to be close to the synthesized single crystal data. This confirmed the formation of small crystallite on the surface of HP film with PZDI treatment.

To dig up into the film's growth and surface/bulk characteristics, we measured XRD patterns (PB/PSA and GIXRD).

To study the strain in the respective device, the stresses were measured according to the $2\theta-\sin^2\psi$ method using $\text{Cu-K}\alpha$ radiation in a RIGAKU Smart Lab X-ray diffractometer and beam/parallel slit analyzer (PB/PSA) optics. The diffracted rays were measured at different angles (tilt angle, $\psi = 10-60$) by fixing a diffraction plane of (012) as shown in adjoining Figure R5 (Figure S17). A lower negative slope was observed for PZDI treatment compared to PEDAI and control devices. This suggests that the surface treatment passivates the defect on the surface or at grain boundaries to some extent which results in lower strain in the film.

Figure R5. XRD spectrum at different tilt angles for the control (a), PEDAI (b), and PZDI (c) devices, respectively. Residual strain extracted from the corresponding HPSC device's diffraction strain data as a function of $\sin^2\psi$.

We have revised the related content on page#11.

Moreover, we also delved into the analysis of residual stress in the respected devices following surface treatment by examination of XRD spectra using the 2θ - $\sin^2\psi$ method,⁴² as detailed in the supporting information (Figure S24). We selected the (012) plane $\sim 31.5^\circ$ as the focal point for analysis due to its ability to provide grain information and its diluting orientation effect in the linear relationship of 2θ - $\sin^2\psi$. The data illustrate how the scattering peaks progressively shift to the left as the ψ angle varies from 10 to 60° at different levels. The stress induced in perovskite can be calculated by fitting the plots (2θ - $\sin^2\psi$). Notably, a negative slope indicates the presence of tensile stress within the perovskite film. A lower negative slope was observed for PZDI treatment compared to PEDAI and control device which aligns with the other reports.^{42,43} This suggests that the surface treatment passivates the defect on the surface or at grain boundaries to some extent which results in lower strain in the film. Thus, the surface treatment using PZDI with stronger -NH functionality regulates the HP film by mitigating the residual strain which is crucial in influencing carrier dynamics and device performance as observed in this work.

To differentiate the effect of additives on the bulk and surface of the film, we collected GIXRD ($\omega = 0.05, 0.1, 0.2 \dots 3^\circ$) of respective films (Figure R6). As with strain analysis, we selected the (012) plane at ~ 31.4 - 31.8° as used for strain analysis due to its ability to provide grain information. One can see the lowest intensity at $\omega = 0.05^\circ$ which is surface sensitive while $\omega = 3^\circ$ could penetrate more than 600 nm which is bulk sensitive. It is found that with the increase in grazing angle, the characteristic XRD peak gradually shifts. The shifting at a low grazing angle ($\omega > 0.5^\circ$) is much more significant for the control sample compared to the surface-treated HP film. As displayed in Figure R6g, the XRD characteristic corresponding to lower penetration depth primarily stems from the film depth of 5-30 nm. One can notice a distinct effect on the surface and bulk of perovskite film. This data suggests that the film surface and grain boundary characteristics have been modified by surface treatment on the HP film. This could have a crucial role in strain attenuation and defect passivation. This observation aligns with device results and device stability.

Figure R6. GIXRD spectra: the control (a, d), PEDAI (b, e), and PZDI (c, f). The plot of penetration depth corresponds to the grazing incident angle. The X-ray attenuation length (penetration depth) into the perovskite film (estimated density $\sim 3.86 \text{ gcm}^{-3}$) was calculated using a report by Davis and co-workers (Atomic Data and Nuclear Data Tables, 1993, 54, 181-342) and Rigaku-manual.

Comments #R2-3

3) Regarding the post-surface treatment, the condition (concentration) and method of the solution should be critical in terms of reproducibility. The use of IPA solution could dissolve the surface of the 3D-perovskite layer as well as PbI_2 and work to form a 2D-like layer at the surface. The authors should give the details more clearly.

We thank the reviewers for their comments.

So far, we have not noticed any dissolving issue with IPA. We found that the surface treatment interacts with remanent PbI_2 to form 2D-layer perovskite as shown in supporting information (XRD given in supporting Figure S1ab and Figure S3).

Noting the reviewer's comment, we have included detailed information about surface treatment in the experiment section.

We also have included the effect of concentration of PEDAI and PZDI in Figures R7 and 8 (device parameters and statistics, device results: Figure S7 and S8, Table S1 and S2, in supporting information).

Figure R7: Figure S7. Statistics of PV characteristic parameters of control and PEDAI treated HPSCs, including V_{oc} , J_{sc} , FF, and PCE. These data consist of 30 devices from 4 batches.

Figure R8: Statistics of PV characteristic parameters of control and PZDI-treated HPSCs, including V_{oc} , J_{sc} , FF, and PCE. These data consist of 30 devices from 4 batches.

Reviewer #3 (Remarks to the Author):

In this manuscript, the authors improved the performance of inverted PSCs by top surface passivation employing piperazine dihydriodide which possesses an alkyl ring structure. The molecular possesses two –NH groups, which can passivate the surface defect, meanwhile promoting the carrier extraction through surface chemistry and band energy modification. The authors extensively studied the working mechanism from the viewpoint of structural, optical, and electrical effects, and demonstrated the universality in wide bandgap and narrow bandgap PSCs.

However, plenty of molecules containing the ammonia group have been employed and depicted effectiveness for surface passivation, hence the manuscript exhibited less novelty, especially considering the Nature Communications is a multidisciplinary Journal. The current manuscript is more suitable to be published in Journals focusing on energy or materials.

Several suggestions also listed below:

We are thankful to the reviewer for the positive comment on the manuscript content.

As noted by the reviewer, we have used diammonium di-iodide functional molecules as surface passivators. As for the question of novelty, as we see two functional molecules used in this report, we have also considered the effect of functional molecules with alky or aryl core.

Importantly, unlike other reports, we have investigated the effect of surface charge-modulated molecular bonding for defect passivation using two distinct functional molecules in the eye of the materials chemistry and device physics. These effects (in film quality, device performance, and stability) have been thoroughly investigated and validated with DFT calculations with detailed insight.

We hope that the revisions made to the manuscript will elicit a favorable response from the esteemed reviewer.

Comments #R3-1

1) The champion PCE should be updated.

We have updated the champion PCE in the main text and added related references.

Comments #R3-2

2) PEDAI is easy for 2D material formation, while PZDI prefers to cover on the grain surface. In addition, PEDAI is easier to penetrate into perovskite, while which has a similar size to PZDI. The detailed mechanism should be explained.

We thank the reviewer for the insightful comment.

It is interesting to note the characteristic difference of PEDAI and PZDI (Figures R9 and R10). We have explained their characteristic difference considering their basicity, charge disparity, and adsorption potential on perovskite surface (as discussed in DFT analysis).

We also have included the size of the respective molecules in the revised figure as displayed below. The PEDAI size varies from 10.39 - 8.50 Å and the size for PZDI is 7.32 Å. These molecules have stark charge differences. They also exhibit different preferences for forming a chain (PEDAI prefers along the (100) direction and PZDI molecules form a chain along the (010) direction). These characteristics have played a crucial role in the distribution of PEDAI or PZDI in the HP film. On the other hand, the change in Gibbs free energy upon adsorption demonstrates that PZDI molecules bind considerably stronger to the surface with I_{Pb} antisite defect in comparison with PEDAI, with the binding energy 1.54 eV per molecule (1.32 eV for PEDAI). This corroborates that PZDI passivates the defective surface with a stronger quenching tendency forming the stable film covering the surface. The PEDAI molecules having (100) preferential direction and less binding energy could be easier for the penetration in the perovskite bulk through the grain boundaries.

In main the text, we have explained to some extent. Page#13 and Page#15

To elucidate the effects of the DIM passivator on the HP film,

In the case of full surface coverage, we accommodated two PEDAI (PZDI) molecules (refer to Figures S20 and S21), with the I atoms of PEDAI (PZDI) adsorbing atop the Pb atoms of the topmost PbI_2 surface layer, as illustrated in Figures S22a, b. Both molecules are assumed a tilted orientation, with their N atoms forming a plane parallel to the perovskite surface. Our calculations revealed that PEDAI molecules exhibit a preference for forming a chain along the (100) direction, while PZDI molecules form a chain along the (010) direction. The interaction of PEDAI and PZDI with the surface induces a slight distortion of the surface PbI_2 layers and causes a rotation of the FA molecules within the first and second PbI_2 -FAI bilayers. This effect is particularly prominent in the case of PEDAI@FAPbI₃ (see Figure S22a). It could also affect the distribution of passivating molecules during the film formation.

.....

Page #15

The observed differences in the distribution tendencies of PEDAI and PZDI in the HP film, as seen in the ToF-SIMS results (Figure 3h-j), can be correlated with their distinct characteristics such as stark charge difference, their preference for forming a chain (PEDAI/PZDI along- (100)/(010) direction) and surface binding energy (1.32/1.54 eV per molecule for PEDAI/PZDI). These characteristics could play a crucial role in the distribution of PEDAI or PZDI in the HP film. The PEDAI molecule having (100) preferential direction and less binding energy promotes the surface coverage as well as the penetration into the bulk, which explains the distinct variations observed in the ToF-SIMS results. Thus, penetration of PEDAI into the perovskite film tends to reduce conductivity, and therefore, one might expect a decrease in device performance. However, since surface passivation is still effective (Figure 5d), an improvement in the V_{OC} is achieved. This explains the observed characteristics of low J_{SC} , low FF, but high V_{oc} , along with enhanced stability compared to the control device. Therefore, the overarching strategy here is to find a species that stays primarily on the surface and penetrates the bulk only through defects on grain boundaries, without significantly compromising crystallinity. Having a similar molecular structure to PEDAI and PZDI, one might anticipate a similar effect. However, we stress that their structural integrity when they are placed at the actual interface matters.

Figure R9: The optimized structures of a free 1,4-phenylenediamine dihydriodide (PEDAI), $C_6H_8N_2 \cdot 2HI$, molecule. (a) The most stable *trans*-isomer structure, and (b) the low-energy *cis*-isomer form. Mulliken charges or Mulliken charges with summed H (iodine and nitrogen atoms) were calculated at the B3LYP/def2TZVP level of theory with the use of Gaussian 09.¹⁷

Free 1,4-phenylenediamine dihydriodide (PEDAI), $C_6H_8N_2 \cdot 2HI$, molecules possess several isomeric forms depending on the position of the HI compounds. The most stable *trans*-isomer is shown in Figure S18a, while its *cis* form (Figure S19b) is only 0.086 eV less stable. When PEDAI adsorbs on the PbI_2 -

terminated surface of FAPbI₃ adsorption of the *cis*-form became energetically favorable, as it maximizes interaction of I atoms of PEDAI with the surface Pb atoms.

Figure R10: The optimized structure of a free piperazine dihydriodide (PZDI), C₄H₁₀N₂*2HI, molecule. Mulliken charges were calculated at the B3LYP/def2TZVP level of theory with the use of Gaussian 09.

Comments #R2-3

3) The better EQE response in $450 > \lambda > 330$ nm is attributed to betterment in interface quality of PZDI-treated HPSC, which is not convincing. As the photon in this part are mainly absorbed around the bottom of the active layer.

We thank the reviewer for this comment.

Indeed, it is complicated to decode the specific range of the EQE spectrum clearly (Figure R11). A quantitative analysis of EQE spectra (Fujiwara and co-workers, *Journal of Applied Physics* 120, 064505 (2016)) has explained the importance of the quality of perovskite surface for EQE response. It is found that the EQE spectra with various interface qualities induced by surface passivation affect the EQE response.

As we compared the dead layer (Figure R11a,b), which is analogous and partially correlated with interface quality, the EQE response (Figure 11c) in this work can be explained.

Accounting for this observation, we believed that the 2D perovskite formed at the interface of ETL and perovskite layer improved the interface quality passivating the

defective layer (it is also supported by TRPL spectra) which results in better EQE response in the lower wavelength regime.

Figure R11: (a, b) Optical models for HPSCs reported by Fujiwara et al. (J. Appl. Phys. 2016,120, 064505) (a, b). This EQE analysis shows the effect of the HTL layer and dead layer (interface layer on the top of 3D-perovskite). (c) it shows the EQE spectra of this report.

We are grateful to the reviewers for carefully reading the manuscript and providing constructive suggestions. We believe that with the changes mentioned, we have addressed the reviewer's concerns.

With these changes and corrections as suggested by the reviewers, we respectfully request that the revised manuscript be reconsidered for publication in *Nature Communications*.

All authors have seen and approved it, and there are no conflicts of interest.

Thank you very much for your kind consideration.

On behalf of all authors,

Sincerely,

Dr. Dhruba B. Khadka

REVIEWER COMMENTS

Reviewer #1 (Remarks to the Author):

The manuscript has been improved a lot during the revision. It could be a suitable candidate for Nature Communications after addressing the following comments.

1. Since the title of the manuscript is "Defect Passivation in Methylammonium/Bromine Free Inverted Perovskite Solar Cells Using

Charge-Modulated Molecular Bonding", the author should make a systematic survey about the MA-free PSCs and cite mile-stone papers in this field, especially the original papers and important recent progresses.

2. The author should also clarify why it is necessary and important to do defect passivation particularly for MA-free perovskite.

Reviewer #2 (Remarks to the Author):

This paper has been revised following the suggestion from reviewers. Although the authors added a sentence with citation of the prior similar works, an important report (ACS Appl. Mater. Interfaces, 2022, 14, 56290), which demonstrates the universal effects of ethylene diammonium salts for Pb perovskite, is arbitrarily omitted. The paper of Adv. Mater. 2023 cited as ref 34, reports the passivation effects of piperadine derivatives, which are very similar to this work. In the revised manuscript, these facts and any new effects beyond these prior reports are not given clearly. Although this paper gives various data on the passivation effects of diammonium salts, it seems difficult to find enough novelty of this work expected for Nat. Commun.

Reviewer #3 (Remarks to the Author):

The authors have adequately addressed concerns of reviewers, and the manuscript quality is improved substantially. Hence it can be considered for publication.

Journal: Nature Communications
Research Article- NCOMMS-23-24660B-Z
Title: Defect Passivation in Methylammonium/Bromine Free Inverted Perovskite Solar Cells Using Charge-Modulated Molecular Bonding

REVIEWER COMMENTS

Reviewer #1 (Remarks to the Author):

The manuscript has been improved a lot during the revision. It could be a suitable candidate for Nature Communications after addressing the following comments.

We sincerely thank the reviewer for the positive comments and for providing us with additional constructive suggestions. We have revised the contents to improve further our manuscript based on the reviewer's suggestions.

The detailed responses to each point are shown below.

Comment #R1-1

1. Since the title of the manuscript is "Defect Passivation in Methylammonium/Bromine Free Inverted Perovskite Solar Cells Using Charge-Modulated Molecular Bonding", the author should make a systematic survey about the MA-free PSCs and cite mile-stone papers in this field, especially the original papers and important recent progresses.

We thank the reviewer for providing constructive suggestions. We have included MA-free PSCs reports noting its progress. In addition to the previously tabulated report (Table S4, also updated), the introduction section has been revised accordingly.

We have thoroughly revised the content by increasing citations from #65 to #84.

On page #2

Methylammonium (MA)-based HPs with a combination of various cations and halides have been extensively used in the state-of-the-art HPSCs.¹⁰⁻¹³ However, MA is prone to be released and decomposes when exposed to elevated temperatures and humid conditions, which poses a persistent concern for device stability.¹⁴⁻¹⁶ In recent years, MA-free formamidinium (FA)-based HP has garnered significant attention due to its higher thermal stability, enhanced moisture resistance, better absorbance in the near-infrared region, and tolerance to varying processing conditions.^{17,18} However, it forms a photoinactive "yellow phase" at room temperature in pristine formamidinium (FA)- perovskite due to the relatively large size of FA. Alkali or organic cations have been introduced into the FA-perovskite film to promote the growth of the photoactive black phase at low-temperature crystallization. For example, Saliba and colleagues have reported the formation of a remarkably crystalline photoactive

phase of FA-HP, achieved by introducing Rb and Cs ions without the use of MA and Br and subjecting the material to annealing temperatures not exceeding 100°C. They have achieved a competitive PCE of 20.35% with improved stability.¹⁹ Similarly, Seok and co-workers demonstrated an impressive PCE of 24.4% using alloyed FA-perovskite with organic cation in the normal device configuration.²⁰ It has documented the formation of a highly crystalline α - phase by incorporating methylenediammonium dichloride as a divalent organic cation with an ionic radius equivalent to FA which induced a stronger ionic interaction of its divalent state in FA-HP.

Comment #R1-2

2. The author should also clarify why it is necessary and important to do defect passivation, particularly for MA-free perovskite.

We thank the reviewer for constructive suggestions. We have revised the introduction section including a discussion on defect passivation reports on MA-free PSCs.

On the pages #2 and 3

In recent scenarios, several studies have focused on the versatile molecular passivation strategy aimed at mitigating various intrinsic defects, whether shallow or deep trap states, within the forbidden region of HP.^{4,21,22} Surface or bulk areas can lead to the formation of under-coordinated Pb^{2+} ions, A-site vacancies, A-site interstitial, and halide vacancies, causing recombination and performance loss.^{22–25} The molecular passivator with multifunctional derivatives consisting of amine,^{26,27} Lewis acids/bases,^{28,29} supramolecules,^{30–32} ionic polymer,^{9,33} etc. have been used for mitigation of defect chemistry in the HPSCs. The ammonium-containing functional additives with alkyl or aryl halide or pseudohalide counterparts have demonstrated a significant enhancement in PCE and operational stability.^{34–41} The additive having stronger adsorption on the HP surface modulates the grain nucleation and growth or alleviates defect chemistry at interface and bulk, which is crucial for a highly efficient and stable device.^{22,42} **Additionally, some functional molecules eliminate molecular iodine present within the perovskite and suppress both iodine and Pb-related deep trap sites.**^{43,44} Importantly, the introduction of molecular functional derivatives of diammonium halide into alkyl or aryl core, serving as additives or passivation, has also exhibited noteworthy enhancements in both the PCE and stability of HPSCs.^{45–49} For example, Wakamiya and co-workers have demonstrated the universality of surface treatment with ethylenediammonium diiodide by wet and dry deposition methods on perovskite surfaces, achieving a decent PCE and device stability in various perovskite systems.^{45,50} Liu and co-workers introduced a highly electronegative fluorine molecule; Cobalt (II) hexafluoro-2,4-pentanedionat for

interface passivation of MA-free HPSCs, resulting in the mitigation of defect chemistries and enhancing hole-transport kinetics.⁵¹ This modification demonstrated a remarkable PCE of 24.64% (normal device structure) as a consequence of strong molecular interaction with the surface charge modulated passivation. Typically, inverted devices tend to exhibit lower efficiency levels compared to conventional structures. Inverted PSCs with an inorganic hole transport layer have gained significant interest due to their potential for superior stability and compatibility with tandem solar cells.^{52,53} An inverted PSC with MA-free HP mixing Tris(chloromethyl) ammonium iodide using NiO_x nanoparticle as HTL reported a decent certified PCE of 23.2% with a small active area (0.04 cm²).⁴² Despite achieving an impressive PCE of record level in small-area PSCs, there remains a substantial PCE disparity between small and large-area PSC devices.^{4,54} Therefore, the fabrication of highly efficient inverted PSC with a large area continues to pose a challenge. Despite a variety of functional molecules being wielded for the optimization of crystal growth and defect passivation in MA-free HP, the influence of adjusting surface charge via bonding interactions is often overlooked.

Reviewer #2 (Remarks to the Author):

This paper has been revised following the suggestions from reviewers. Although the authors added a sentence with citation of the prior similar works, an important report (ACS Appl. Mater. Interfaces, 2022, 14, 56290), which demonstrates the universal effects of ethylene diammonium salts for Pb perovskite, is arbitrarily omitted.

The paper of Adv. Mater. 2023 cited as ref 34, reports the passivation effects of piperadine derivatives, which are very similar to this work. In the revised manuscript, these facts and any new effects beyond these prior reports are not given clearly. Although this paper gives various data on the passivation effects of diammonium salts, it seems difficult to find enough novelty of this work expected for Nat. Commun.

We sincerely thank the reviewer for the concern about citing the previous reports and the novelty of our report.

We have split the comments into two parts for ease of Author's reply

Comment #R2-1

This paper has been revised following the suggestions from reviewers. Although the authors added a sentence with a citation of the prior similar works, an important report (ACS Appl. Mater. Interfaces, 2022, 14, 56290), which demonstrates the universal effects of ethylene diammonium salts for Pb perovskite, is arbitrarily omitted.

We apologize for missing the noted report (*Wakamiya and co-workers, ACS Appl. Mater. Interfaces, 2022, 14, 56290*) in the revised manuscript. We highly appreciate the reviewer's comment and sharing interesting reports related to the diammonium salt.

As per the reviewer's suggestion, we have included related reports and thoroughly revised the introduction section to enrich the background content. We also have referred to these reports to support our experimental results.

We have revised the content in the introduction section or results and discussion accordingly.

On the pages #2 and 3

In recent scenarios, several studies have focused on the versatile molecular passivation strategy aimed at mitigating various intrinsic defects, whether shallow or deep trap states, within the forbidden region of perovskite materials.^{4,21,22} Surface or bulk areas can lead to the formation of under-coordinated Pb²⁺ ions, A-site vacancies, A-site interstitial, and halide vacancies, causing recombination and performance loss.²²⁻²⁵ Additionally, some functional molecules eliminate molecular iodine present within the perovskite and suppress both iodine and Pb-related deep trap sites.^{41,42} Importantly, the introduction of molecular functional derivatives of diammonium halide into alkyl or aryl core, serving as additives or passivation, has also exhibited noteworthy enhancements in both the PCE and stability of HPSCs.⁴³⁻⁴⁷ For example, Wakamiya and co-workers have demonstrated the universality of surface treatment with ethylenediammonium diiodide by wet and dry deposition methods on perovskite surfaces, achieving a decent PCE and device stability in various perovskite systems.^{43,48} Liu and co-workers introduced a highly electronegative fluorine molecule; Cobalt (II) hexafluoro-2,4-pentanedionat for interface passivation of MA-free HPSCs, resulting in the mitigation of defect chemistries and enhancing hole-transport kinetics.⁴⁹ Despite a variety of functional molecules being wielded for the optimization of crystal growth and defect passivation in MA-free HP, the influence of adjusting surface charge via bonding interactions is often overlooked.

On pages #11/12

The characteristic signals from PEDAI or PZDI are found to be significantly higher on the surface with a deep gradient to the bulk (Figure S16). The 3D maps (Figure 3h-j) demonstrate that the PEDAI²⁺ and

PZD²⁺ cations introduced by surface treatment are mainly distributed on the HP top surface. This observation is analogous to the report by the Wakamiya group in ethylenediammonium iodide-treated Sn-Pb mixed HP film.⁴²

On the page#15

The device efficiency of NB-HPSCs using Sn/Pb binary HP materials with reduced MA is in the competitive range of another report.⁶⁸ The V_{OC} deficit of NB-HPSCs is significantly lowered from 0.478 to 0.405 V (Table 2) which is attributed to attenuation of surface or bulk recombination. This result further confirms the effectiveness of PZDI treatment for NB-HPSCs, consolidating both experimental results and theoretical observations. Importantly, this report corroborates the universality of surface treatment using bifunctional diammonium molecules for the enhancement of device PCE and stability by mitigating detrimental defects by modifying the surface and bulk defect chemistry of perovskite film. This observation is parallel to other reports of similar molecular derivatives.^{42,45-47,69}

Comment #R2-2

The paper of Adv. Mater. 2023 cited as ref 34, reports the passivation effects of piperadine derivatives, which are very similar to this work. In the revised manuscript, these facts and any new effects beyond these prior reports are not given clearly. Although this paper gives various data on the passivation effects of diammonium salts, it seems difficult to find enough novelty of this work expected for Nat. Commun.

A report (Wakamiya and co-workers, Adv. Mater. 2023, 35, 2208320) noted by Reviewer#2 has used piperadine derivatives in Sn-Pb PSCs. This is a very interesting and noteworthy report on perovskite research. They have discussed the effect of the carbon chain in piperadine and optimized the device by surface treatment with mixed precursor PP (piperadine derivatives) + CPTA (C₆₀ functional derivative).

Although the surface treatment method is similar to this report, we think the molecular functionalities behind the surface treatment and characteristic insights are significantly different from our report.

We believe that our report unraveled a new effect of charge-modulated molecular passivation; PEDAI and PZDI in perovskite solar cells.

In this work, we have investigated the passivation strategy through bond/charge-regulated defect passivation by introducing bifunctional molecules with an aryl core (1,4-phenylenediamine dihydride (PEDAI)) or alkyl core (piperazine dihydride (PZDI)) onto the MA/Br-free perovskite. To the best of our knowledge, there is no such comparative discussion reported in PSC.

Although PEDAI and PZDI molecules share chemical similarities, this work revealed discernible effects on film growth, material distribution, and device characteristics. These outcomes were extensively investigated through a combination of experimental data and theoretical calculations.

Our study delved into a comprehensive analysis of molecular passivation using PEDAI and PZDI, drawing comparisons between their molecular functionalities. We contend that our work has uncovered a novel aspect related to charge-modulated aryl and alkyl bifunctional molecules; PEDAI and PZDI in perovskite solar cells.

Reviewer #3 (Remarks to the Author):

The authors have adequately addressed the concerns of reviewers, and the manuscript quality has improved substantially. Hence it can be considered for publication.

We heartily appreciate the reviewer's positive comment on the revised manuscript. We are heartily thankful to the reviewer for the recommendation.

We are grateful to the reviewers for carefully reading the manuscript and providing constructive suggestions.

We believe that with the changes mentioned, we have addressed the reviewer's concerns.

On behalf of all authors,

Sincerely yours,

Dr. Dhruba B. Khadka